# Diverse fates of uracilated HIV-1 DNA during infection of myeloid lineage cells

Erik C Hansen[1], Monica Ransom[2], Jay R Hesselberth[2], Nina N Hosmane[3], Adam A Capoferri[3,4], Katherine M Bruner[3], Ross A Pollack[3], Hao Zhang[5], Michael Bradley Drummond[3], Janet M Siliciano[3], Robert Siliciano[3,4], James T Stivers[1]*

[1]Department of Pharmacology and Molecular Sciences, The Johns Hopkins University School of Medicine, Baltimore, United States; [2]Department of Biochemistry and Molecular Genetics, University of Colorado School of Medicine, Aurora, United States; [3]Department of Medicine, The Johns Hopkins University School of Medicine, Baltimore, United States; [4]Howard Hughes Medical Institute, The Johns Hopkins School of Medicine, Baltimore, United States; [5]W Harry Feinstone Department of Molecular Microbiology and Immunology, Johns Hopkins Bloomberg School of Public Health, Baltimore, United States

*For correspondence: jstivers@jhmi.edu

**Abstract** We report that a major subpopulation of monocyte-derived macrophages (MDMs) contains high levels of dUTP, which is incorporated into HIV-1 DNA during reverse transcription (U/A pairs), resulting in pre-integration restriction and post-integration mutagenesis. After entering the nucleus, uracilated viral DNA products are degraded by the uracil base excision repair (UBER) machinery with less than 1% of the uracilated DNA successfully integrating. Although uracilated proviral DNA showed few mutations, the viral genomic RNA was highly mutated, suggesting that errors occur during transcription. Viral DNA isolated from blood monocytes and alveolar macrophages (but not T cells) of drug-suppressed HIV-infected individuals also contained abundant uracils. The presence of viral uracils in short-lived monocytes suggests their recent infection through contact with virus producing cells in a tissue reservoir. These findings reveal new elements of a viral defense mechanism involving host UBER that may be relevant to the establishment and persistence of HIV-1 infection.

## Introduction

The uracil nucleobase plays a central role in adaptive and innate immunity against HIV-1 when it is found in DNA rather than RNA (*Priet et al., 2006*; *Sire et al., 2008*). The most well-characterized uracil-centric innate immune response involves host cell DNA cytidine deaminase enzymes (APO-BECs), which selectively deaminate cytosine residues during (-) strand DNA synthesis thereby rendering the viral genome nonfunctional by hypermutation (C/G to T/A). Work from our lab and others has suggested the presence of another uracil-mediated HIV-1 restriction pathway involving the incorporation of dUTP into viral DNA by reverse transcriptase to produce U/A base pairs ('uracilation') (*Weil et al., 2013*). The dUTP-dependent pathway is thought to be restricted to non-dividing immune cells such as macrophages, monocytes and resting CD4+ T cells because only non-dividing cells have the requisite low levels of canonical dNTPs and elevated ratios of dUTP/ TTP (*Yan et al., 2011*; *Kennedy et al., 2011*). Notably, U/A pairs resulting from dUTP incorporation are 'invisible' to normal DNA sequencing methods and they retain the coding potential of normal T/A base pairs. Despite the similarity of U/A and T/A pairs, the presence of uracil in DNA has the potential to introduce diverse effects on viral infection including transcriptional silencing and engagement of the host

**eLife digest** Human immunodeficiency virus type 1 (HIV-1) infects and kills immune cells known as CD4+ T cells, leading to the disease AIDS. Current drug treatments enable HIV-1 infected patients to live relatively long and healthy lives. However, no cure for HIV-1 exists because the virus lives indefinitely in a resting state within the genetic material – or genome – of the infected cell, where it is not susceptible to drug treatments. Most HIV-1 research focuses on T cells, but another type of immune cell – the macrophage – may also harbor resting HIV-1 in its genome.

Compared to other cells, macrophages are unusual because they produce large amounts of a molecule called deoxyuridine triphosphate (dUTP). Most cells, including T cells, keep dUTP levels very low because it closely resembles molecules that are used to make DNA and so it can be accidentally incorporated into the cell's DNA. When this happens, the cell removes the dUTP from the DNA using enzymes in a process called uracil base excision repair (UBER). To hide inside the cell's genome, HIV-1 needs to produce a DNA copy of its own genome, but it was not known what happens when HIV-1 tries to do this within a macrophage that contains high levels of dUTP and UBER enzymes.

Here, Hansen et al. reveal that about 90% of macrophages have exceptionally high levels of dUTP and are poorly infected by HIV-1. The high levels of dUTP result in the virus incorporating dUTP into its DNA, which is then attacked and fragmented by UBER enzymes. However, about one in a hundred viral DNA molecules do manage to successfully integrate into the genome of the macrophage. This viral DNA later gives rise to new virus particles through an error-prone process that, by introducing new mutations into the virus genome, may help HIV-1 to evolve and persist.

Further experiments examined cells that give rise to macrophages from infected patients who had been on anti-HIV drug therapy for several years. Hansen et al. found that there was lots of dUTP in the DNA sequences of HIV-1 viruses found in these "precursor" cells. These precursor cells only live for several days before being eliminated, so the presence of viruses containing dUTP suggests these cells were infected recently. A future challenge will be to identify new anti-HIV drugs that specifically target macrophages and to understand the role of error-prone production of new viral genomes.

uracil base excision repair (UBER) pathway (*Brégeon et al., 2003*; *Krokan et al., 2013a*, *2014b*). The presence, persistence, and ultimate fate of U/A pairs in HIV-1 proviral DNA is unexplored and is important for understanding the potential of tissue macrophages to establish and maintain a long-term HIV reservoir.

A role for dUTP and the UBER enzyme uracil DNA glycosylase (hUNG2) in HIV-1 infection have been long debated [see reference (*Weil et al. 2013*) for a succinct review]. A dUTP-mediated host defense pathway was first suggested from inspection of the genomes of β-retroviruses, non-primate lentiviruses, and endogenous retroviruses, which have all captured a host dUTPase gene during viral evolution. Additional support was provided by the observation that dUTPase-deficient mutants of non-primate lentiviruses cannot infect non-dividing cells that contain high-dUTP levels (*Lichtenstein et al., 1995*; *Turelli et al., 1996*; *Threadgill et al., 1993*; *Steagall et al., 1995*). Intriguingly, HIV-1 does not encode a dUTPase yet can infect human macrophages with a reported dUTP:TTP ratio of 60:1 (*Kennedy et al., 2011*). There is an equal level of intrigue concerning the role of hUNG2 in HIV infection. A *pro-infective* role for hUNG2 has been suggested by reports that hUNG2 suppresses mutations in the viral genome upon infection of macrophages (*Mansky et al., 2000*; *Chen et al., 2004*; *Priet et al., 2005*; *Guenzel et al., 2012*), but is completely dispensable for HIV-1 replication of cells with low-dUTP levels (*Kaiseer and Emerman, 2006*). In contrast, a modest *restrictive* role for hUNG2 has been suggested from the decreased infectivity of HIV virions lacking viral protein R (Vpr). This restriction is attributed to a Vpr-dependent ubiquitin-mediated hUNG2 degradation pathway or through Vpr-induced transcriptional silencing of hUNG2 expression (*Schrofelbauer et al., 2005*; *Ahn et al., 2010*; *Langevin et al., 2009*). These intriguing prior observations have motivated our further studies into the role of UBER in HIV infection, which now establish a profoundly restrictive role and unexpected effects on viral mutagenesis.

## Results

### Unique nucleotide metabolism in myeloid cells results in high dUTP/TTP

We hypothesized that viral uracilation and restriction in resting immune cells would require enzyme activities that support a high dUTP/TTP ratio and uracil base excision. Using sensitive and specific in vitro enzymatic assays (*Figure 1—figure supplement 1A–D*) (*Weil et al., 2013*; *Hansen et al., 2014*; *Seiple et al., 2008*), we found that monocytes and monocyte-derived macrophages (MDMs) expressed high levels of SAMHD1 dNTP triphosphohydrolase to reduce the canonical dNTP pools (*Hansen et al., 2014*; *Goldstone et al., 2011*), undetectable dUTPase activity that allowed dUTP accumulation, and modest expression of the UBER enzymes uracil DNA glycosylase (hUNG) and abasic site endonuclease (APE1) (*Figure 1—figure supplement 1E–H*). Although resting CD4$^+$ T cells also possessed high SAMHD1, hUNG and APE activities, their dUTPase activity was at least seven-fold greater than MDMs. LC-MS analyses of the dUTP and canonical dNTP levels in resting and activated CD4$^+$ T cells and MDMs revealed that the dUTP/TTP ratio was ~20 for MDMs, 1.1 for resting CD4$^+$ T cells, and <0.05 for activated CD4$^+$ T cells (*Figure 1—figure supplement 1I,J*) (*Gavegnano et al., 2012*; *Hollenbaugh et al., 2014*). Since reverse transcriptase has a nearly identical $K_m$ for dUTP and TTP (*Kennedy et al., 2011*), the high ratio of dUTP/TTP indicates that uracil incorporation is a very frequent event during reverse transcription in the cell cytoplasm, which contains no UBER activity.

### Total HIV-1 DNA isolated from in vitro infected MDMs is heavily uracilated

To detect and map uracil in selected regions of the HIV genome, we developed uracil excision-droplet digital PCR (Ex-ddPCR) (*Figure 1A*). Briefly, Ex-ddPCR involves isolation of total DNA from HIV-infected MDMs, after which half of the sample is treated with UNG to destroy template strands that contain uracil (*Figure 1A*). Thus, any PCR amplicon that contains one or more uracils on each DNA strand is not amplified in the UNG-treated DNA sample. After performing ddPCR (no UNG treatment) and Ex-ddPCR (UNG pre-treatment), the fraction of the amplicons containing at least one uracil on each strand was calculated from the counts of positive droplets for the ddPCR and Ex-ddPCR samples (*Figure 1A*). A complete description of Ex-ddPCR is found in Materials and methods and *Figure 1—figure supplement 2A–E*.

Ex-ddPCR analysis of HIV DNA isolated from single-round infections of activated and resting CD4 + T cells and MDMs using a VSV-G pseudo-typed replication-deficient HIV strain (HIV$^{NL4.3(VSVG)}$) established that uracilation of viral DNA occurs in MDMs, but not T cells (*Figure 1B*). The scatter plots and histograms in *Figure 1B* show that the copy number for viral *gag* DNA isolated from MDMs at 3 days post-infection was 3.5-fold lower in the Ex-ddPCR experiment, showing that ~70% of *gag* amplicons contained uracil. Importantly, the genomic reference standard RNase P (*RPP30*) contained no measurable uracil, indicating that uracil incorporation is specific to HIV DNA and occurs during reverse transcription. In contrast, the *gag* copy number for viral DNA collected from activated and resting T cells was the same for both the ddPCR and Ex-ddPCR reactions indicating that uracil was absent in viral DNA isolated from infected T cells (*Figure 1B*).

To augment Ex-ddPCR, we also applied the next-generation sequencing technology uracil Ex-Seq to globally map the frequency of U/A pairs across the entire HIV genome (*Bryan et al., 2014*). Ex-Seq is similar to standard Illumina sequencing (Seq), except that UNG-mediated uracil excision is used to destroy uracil-containing templates prior to PCR amplification. To specifically enrich HIV-1 sequences, we used 5'-biotin-conjugated DNA probes that tiled both strands of the entire viral genome, yielding a $10^3$-fold increase in HIV-derived fragments. Sequencing of viral DNA isolated from MDMs 7 days after infection with HIV$^{NL4.3(VSVG)}$ showed uniform coverage across the genome except for notably increased reads at the 5′ and 3′-LTR regions, which was equally evident for both the Seq and Ex-Seq samples (*Figure 1C*). We speculate that the elevated signal in the LTRs arises from non-uniform hybridization of the lock-down probes, which is normalized when converting the reads to Frac U. The ratio of the normalized sequencing reads (Ex-Seq/Seq) thus quantifies the fraction that contained at least one uracil on each strand (*Figure 1D*). Thus, about 60% of the 100 bp reads contained uracil, leading us to conclude that uracilation was uniform across the viral genome.

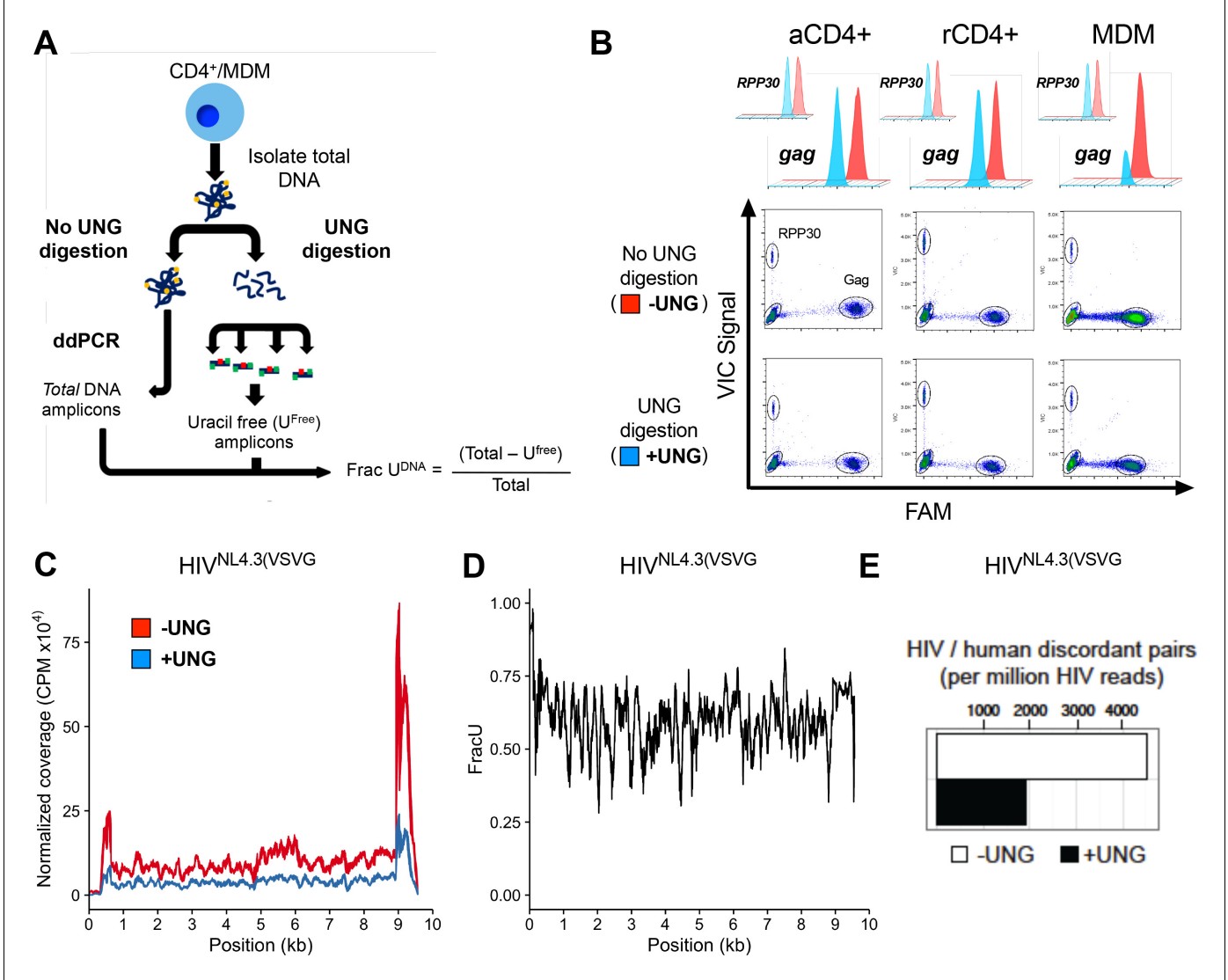

**Figure 1.** Ex-ddPCR and Ex-Seq determine the uracil content of the HIV DNA copies. (**A**) After infection of immune cells with the single-round VSV-G pseudo-typed virus HIV[NL4.3(VSVG)], total DNA is isolated and either digested by UNG to degrade uracil containing DNA or mock digested to provide a measure of total amplifiable DNA in the sample. The output (Frac U[DNA]) represents the fraction of amplified DNA copies containing at least one uracil on each DNA strand. Signal is normalized to a genomic reference copy standard (*RPP30*) that does not contain uracil. (**B**) Activated, resting CD4+ T cells and MDMs were infected in vitro with HIV[NL4.3(VSVG)] virus and the uracil content was measured 3 days post-infection (dpi) using primers that targeted the *gag* region. The data (± UNG digestion) are shown as scatter plots and histograms. (**C**) Normalized coverage of the HIV[NL4.3(VSVG)]-genome-positive strand in Excision-Seq (Ex-Seq) libraries prepared from total cellular DNA at 7 days post-HIV infection. (**D**) Fraction of the reads in panel C that contained uracil (Frac U). (**E**) Discordant read pairs between HIV and human DNA present in Ex-Seq libraries prepared from total cellular DNA at 7 days post-infection with HIV[NL4.3(VSVG)] virus. The number of discordant reads obtained by Ex-Seq in the absence and presence of UNG digestion are shown as white and black bars.

The following figure supplements are available for figure 1:

**Figure supplement 1.** Profiling enzyme activities and dNTP pool levels in immune target cells of HIV.

**Figure supplement 2.** In vitro generated calibration curves for evaluating uracil content in DNA amplicons and single-round HIV infections of cultured MDMs.

## Integrated proviruses contain abundant uracils

The above analyses report on both integrated and unintegrated viral DNA. However, the Ex-Seq experiment also provides specific information on the uracilation status of integrated proviruses and their genomic sites of integration (*Figure 1E*). Sequence reads from integrated proviruses are unambiguously assigned by the presence of viral LTR and human genomic sequences ('discordant pairs'). Over 4000 discordant pairs were identified per million HIV reads for the Seq sample, which was reduced by about 60% for the Ex-Seq sample (*Figure 1E*). This level of uracilation is similar to that of the total HIV DNA and supports our previous finding that uracils can persist in proviruses when nuclear hUNG2 expression is low (*Weil et al., 2013*).

## Uracils arise from high dUTP/TTP not APOBEC-catalyzed cytosine deamination

For T cell targets of HIV-1, expression of APOBEC3G (A3G) and APOBEC3F (A3F) has been shown to result in cytosine deamination at preferred sequence motifs present in the HIV minus-strand DNA, giving rise to mutagenic uracils (C→T transition mutations) (*Harris et al, 2003a*, *2003b*). We excluded APOBEC-catalyzed cytidine deamination as a major source of viral uracils in infected MDMs using several criteria. First, Ex-Seq data showed that the uracils appeared on *both* strands of the viral cDNA rather than the (-) strand that is exclusively targeted by APOBECs (*Yu et al., 2004*). Second, the mutational spectrum of proviral DNA did not match the signature C/G→T/A hotspots for APOBEC cytosine deamination as determined using the program *Hypermut* 2.0 (see sequencing studies below) (*Rose and Korber, 2000*). Finally, we obtained direct evidence that viral uracils originated from the high dUTP/TTP ratio by increasing the intracellular TTP levels through the addition of 5 mM thymidine to the culture media. Addition of thymidine to the culture medium prior to infection led to a 14-fold decrease in the uracilation level of proviral DNA (*Figure 2A*) and a five-fold increase in proviral copies as compared to no thymidine supplementation (*Figure 2B*). The combined results strongly support a mechanism where the viral uracils originate primarily from dUTP incorporation during reverse transcription. Since reverse transcriptase utilizes dUTP with the same efficiency and fidelity as TTP (*Kennedy et al., 2011*), the uracils must be in the form of coding U/A base pairs. In *Figure 2—figure supplement 1*, we present immunoblots showing that MDMs express very low levels of A3G and A3A, which is consistent with previous reports and the absence of tell-tale marks of enzymatic deamination described above (*Harris and Liddament, 2004*).

## Identification of restrictive and permissive MDM subpopulations

During HIV infection of MDMs ~10% of the infected MDMs were positive for GFP fluorescence after infection with either replication deficient HIV[NL4.3(VSVG)] or replication competent HIV[SF162(CCR5)] (*Figure 3A*). This level of infection based on GFP fluorescence remained constant from about seven to 30 days after infection and did not change when the MOI was varied in the range 0.1–10 (*Figure 3—figure supplement 1*). This result led us to suspect that the MDM population might consist of a phenotypic mixture with different susceptibilities to HIV infection. (We note that higher MOIs are typically used in our experiments to obtain reasonable viral copy numbers in the GFP⁻ population, but the results are independent of MOI.) Accordingly, we used GFP fluorescence to sort HIV[NL4.3(VSVG)] infected MDMs into 99% pure GFP positive and negative populations (*Figure 3—figure supplement 2*). Droplet digital PCR measurements showed that the copy number of early viral *gag*-containing DNA products in the GFP⁻ population was about seven-fold lower than the GFP⁺ population at 1 day post-infection (~2 versus 14 copies/cell, *Figure 3B*). This result suggested that viral infection was hindered at the entry and/or early reverse transcription steps in the GFP⁻ population. Using *ALU-gag* nested qPCR to measure proviral copies (*O'Doherty et al., 2002*), we found that the GFP- population contained only 1% of the copies seen in the GFP⁺ MDMs (*Figure 3C*). The large decrease in copy number between the early DNA intermediates and the provirus stage suggested that a potent pre-integration restriction mechanism was also present in the GFP⁻ population.

## Viral uracilation is exclusive to GFP⁻ MDMs

We used infections with a replication-competent and macrophage-tropic virus (HIV[SF162(CCR5)]) to further elucidate the characteristics of the two MDM populations. First, the infected MDMs were sorted into GFP positive and negative populations at 1 day post-infection. The cells were then cultured at

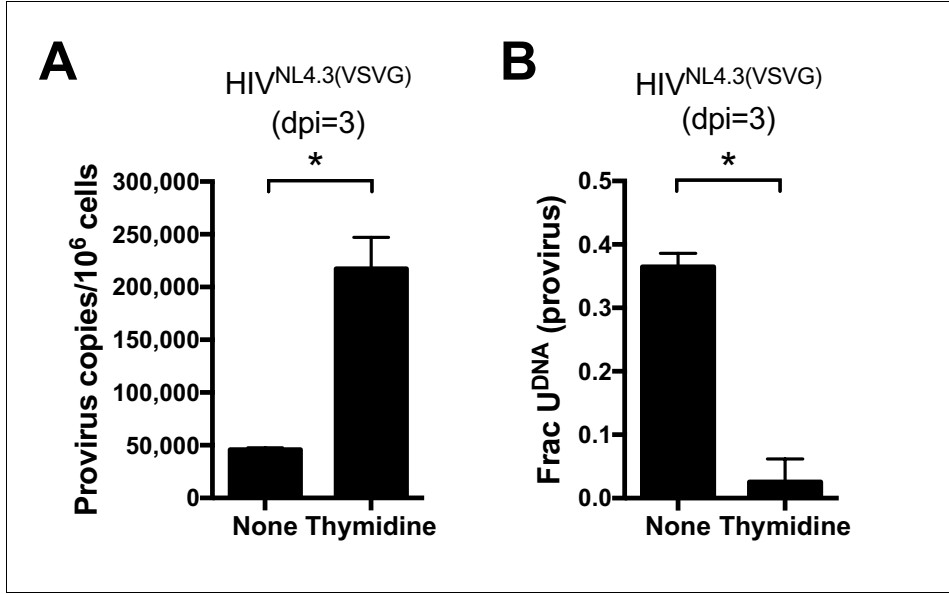

**Figure 2.** Uracils arise from high dUTP/TTP not APOBEC-catalyzed cytosine deamination. (**A**) Culturing a mixed GFP+/GFP− MDM cell population with media supplemented with 5 mM thymidine resulted in a 14-fold decrease in the fraction of proviral copies that contained uracil. Uracil content was measured using Ex-*ALU-gag* nested qPCR with their statistical significance level (*p<0.05). (**B**) Mixed MDM cultures supplemented with 5 mM thymidine showed a five-fold increase in provirus copy number as measured using *ALU-gag* nested qPCR. Number of experimental replicates (n = 4–5) and errors are reported as mean ± SD.

The following source data and figure supplement are available for figure 2:

**Source data 1.** Ex-*ALU-gag* qPCR for provirus and uracil detection in MDMs cultured with standard media or media supplemented with 5 mM Thymidine (*Figure 2A*).

**Figure supplement 1.** APOBEC3G (A3G) and APOBEC3A (A3A) are poorly expressed in a mixed population of MDMs.

---

days 1, 7, 14, and 30 post-sorting where we measured proviral DNA copy number, the fraction of proviruses containing uracil, and the HIV p24 levels in the culture supernatants for each cell population (*Figure 3D,E*). Both MDM populations showed high levels of viral *gag* DNA products at day 1 (red circle and blue square, *Figure 3D*). The GFP+ population was characterized by efficient integration and a high proviral DNA copy number that remained stable for 14 days. By day 30, the copy number increased by ~four-fold, which we attribute to secondary infection by released viruses (white bars, *Figure 3D*). In contrast, the GFP− population had a low efficiency of integration exemplified by a 50-fold reduction in copy number between the early *gag* DNA products and proviruses (black bars, *Figure 3D*). Unlike the GFP+ cells, the proviral copy number in the GFP− population decreased by 20-fold between days 7 and 14 before returning to the day 7 level at 30 days. Once again, we attributed the later increase to secondary infection by released virus particles because it was not observed in single-round infections. We established that HIV^NL4.3(VSVG), HIV^SF162(CCR5), and HIV^BAL ^(CCR5) viral strains and MDMs isolated from different donors behaved similarly (*Figure 3—figure supplement 3A–E*), and that the results were independent of whether differentiated MDMs were obtained using adherence, M-CSF or GM-CSF protocols (*Figure 3—figure supplement 3F–H*).

The most striking difference between the two MDM populations was that the minor GFP+ population had no detectable proviral uracils, while ~90% of the proviral copies in the major GFP− fraction contained uracil at day one (*Figure 3D*). The fraction of proviral copies that were uracilated (Frac U) decreased from 0.77 to 0.23 between days 7 and 14 before rising to 0.55 at day 30 (presumably due to continuing infection). We explored the mechanistic basis for the differences in viral uracilation between the two MDM populations by measuring the dUTP/TTP ratio in the cell extracts obtained

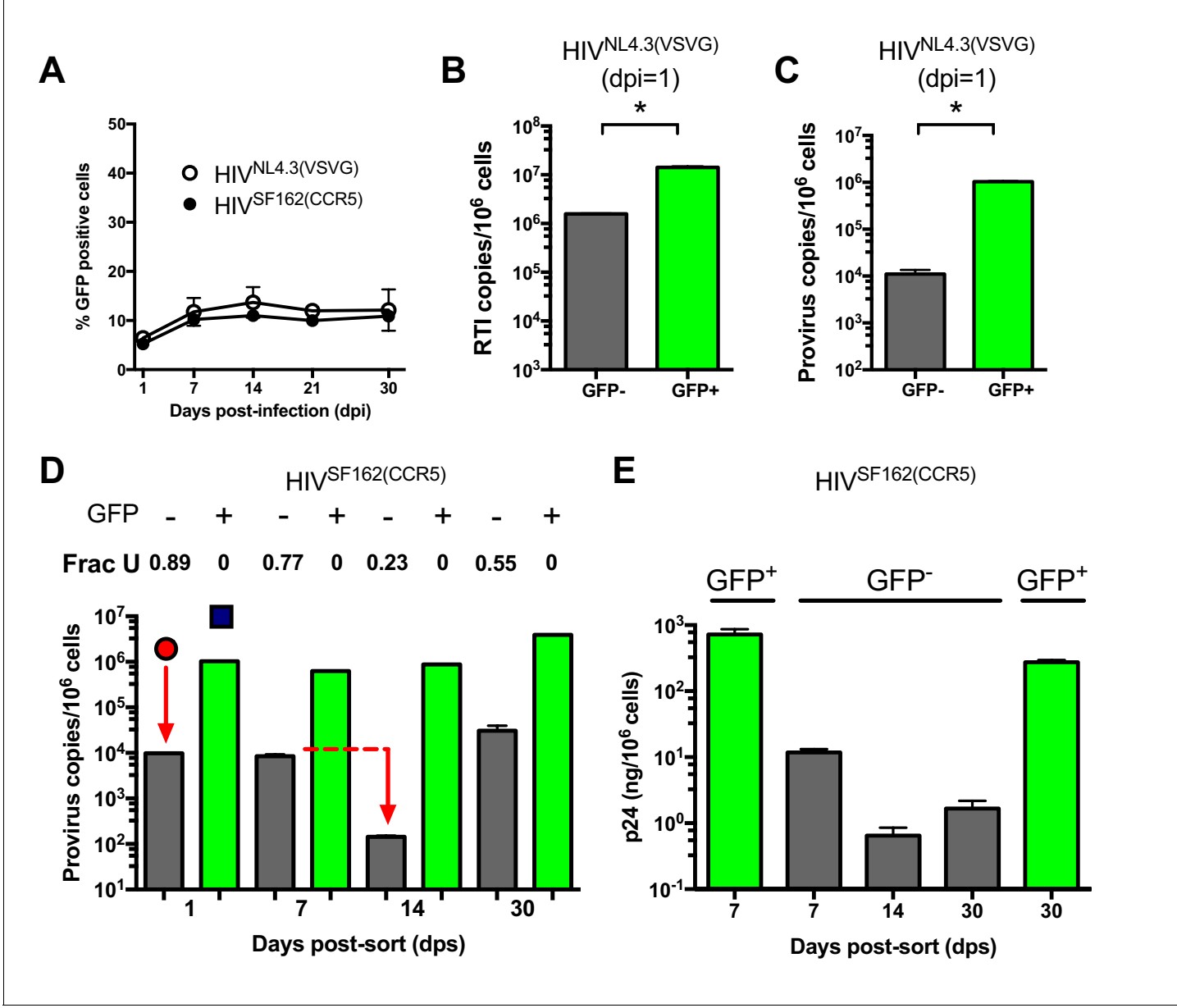

**Figure 3.** MDMs consist of two distinct cell populations with respect to viral infection. (A) Flow cytometry analysis of MDMs infected with replication-deficient HIV[NL4.3(VSVG)] and replication comeptent HIV[SF162(CCR5)] viruses. Expression of virally encoded GFP was only observed in 12–15% cells over a 30-day time-period. (B) After cell sorting according to GFP fluorescence, ddPCR was used to measure the copy number of HIV[NL4.3] reverse transcription intermediates (RTIs) in GFP⁻ (white bars) and GFP⁺ (green bars) MDMs. Level of statistical significance (*p<0.05). (C) *ALU-gag* nested qPCR measurement of the HIV[NL4.3(VSVG)] provirus copy number in sorted GFP⁻ and GFP⁺ MDMs. (D) Measurement of HIV[SF162(CCR5)] provirus copy number and uracil content over the course of a 30-day multi-round infection. (E) ELISA measurements of viral p24 protein levels in GFP⁻ and GFP+ MDMs over the course of a 30-day multi-round infection. Number of experimental replicates (n = 3–5) and errors are reported as mean ± SD.

The following source data and figure supplements are available for figure 3:

**Source data 1.** Ex-*ALU-gag* qPCR for measurement of reverse transcription intermediates (RTIs) and provirus content in GFP sorted MDMs (*Figure 3B,C*).
**Figure supplement 1.** Uracilation is independent of multiplicity of infection (MOI).
**Figure supplement 2.** The sorted populations of in vitro infected GFP− and GFP+ MDMs are highly pure.

Figure 3 continued

**Figure supplement 3.** Three different viral strains show a similar uracilation profile with in vitro infected MDMs independent of the differentiation regimen.

**Figure supplement 4.** GFP expression from HIV[NL4.3] infected MDM populations does not strongly correlate with CD14 and CD16 expression.

from each. The GFP[+] MDMs had a 20-fold lower dUTP/TTP ratio, which stemmed in part from a five-fold higher TTP concentration compared to the GFP[−] MDMs. This difference between the GFP+ and negative MDMs confirms the key mechanistic requirement for viral uracilation: a high dUTP/TTP ratio.

Consistent with their larger proviral copy numbers, the GFP[+] MDMs produced 100 to 1000 times more p24 than the GFP[−] MDMs and the levels were fairly constant over a 30 day period (*Figure 3E*). In contrast, the GFP[−] MDMs showed a 10-fold decrease in p24 levels between day seven and 14 (*Figure 3E*), which coincides with the 20-fold reduction in proviral copy number in the same time period (*Figure 3D*). The expression of detectable levels of p24, but not GFP, in the GFP[−] MDMs suggests that truncated viral RNAs are being expressed that extend through the p24 coding sequence, but not GFP.

## CD14/CD16 status of MDMs does not correlate with GFP expression

We were quite surprised that the uracilation phenotype was restricted solely to MDMs that were incapable of expressing virally encoded GFP (*Figure 3A*) and were curious whether the minor, but highly infectable GFP[+] MDM population might be derived from CD16[+] monocyte precursors. Our reasoning is based on the reports that CD16[+] monocytes are enriched in the CCR5 co-receptor and are the preferred target of HIV as compared to the more prevalent CD14[+] monocytes which express lower levels of CCR5 and appreciable levels of the active low-molecular-weight forms of APOBEC3A and APOBEC3G (*Ellery et al., 2007*). In addition, CD16[+] monocytes typically comprise around 10% of the monocyte pool in uninfected donors (*Ellery et al., 2007*), which curiously matches the level of GFP[+] MDMs in our studies. Despite these suggestive relationships, multicolor flow cytometry established that the GFP fluorescence did not track with the CD14[+] or CD16[+] status of the MDMs (*Figure 3—figure supplement 4*). Aside from the 20-fold higher dUTP levels in the GFP[−] MDMs (see above), we have no comprehensive understanding of what additional phenotypic differences exist between the two macrophage populations that result in the combined GFP[−]/uracilation phenotype. It is possible that the dominant GFP[−] phenotype arises from a complex combination of effects related to the metabolic state, or activation state of these cells.

## Viral uracilation is modulated by hUNG2 activity

We previously established that hUNG2 activity was the primary determinant of whether uracilated proviral DNA survived or persisted in the HT29 model cell system (*Weil et al., 2013*). To confirm this result in MDMs, we took advantage of three approaches: (i) knockdown of hUNG2 activity by transfection with a plasmid vector that overexpressed the potent uracil DNA glycosylase inhibitor protein (Ugi) (*Bennett et al., 1993*), (ii) overexpression of hUNG2 by transfection with a plasmid expression vector, and (iii) infections with viral constructs where virus protein R (Vpr) was deleted. Viral protein R is known to interact with hUNG2 and induce its degradation through the assembly with the DDB1-CUL4 ubiquitin ligase complex and would be expected to recapitulate the phenotype produced by Ugi overexpression (*Eldin et al., 2014*). We surmised that reduced hUNG2 levels would lead to greater infection efficiency, higher levels of uracil in viral DNA products, and that greater expression of hUNG2 would correlate with increased restriction and reduced uracil content in proviral DNA due to uracil excision by hUNG2. It is important to note that plasmid DNA transfection methodology is essential in these experiments because the introduction of Ugi and hUNG2 through lentiviral packaging and transfection would be compromised by uracilation during reverse transcription.

Primary macrophages derived from blood monocytes were cultured in the presence of M-CSF for 7 days prior to transfection with the pIRES-Ugi-eGFP and pIRES-hUNG2-eGFP expression plasmids (*Figure 4A*). Because MDMs are exceedingly difficult to transfect, the cells expressing eGFP were

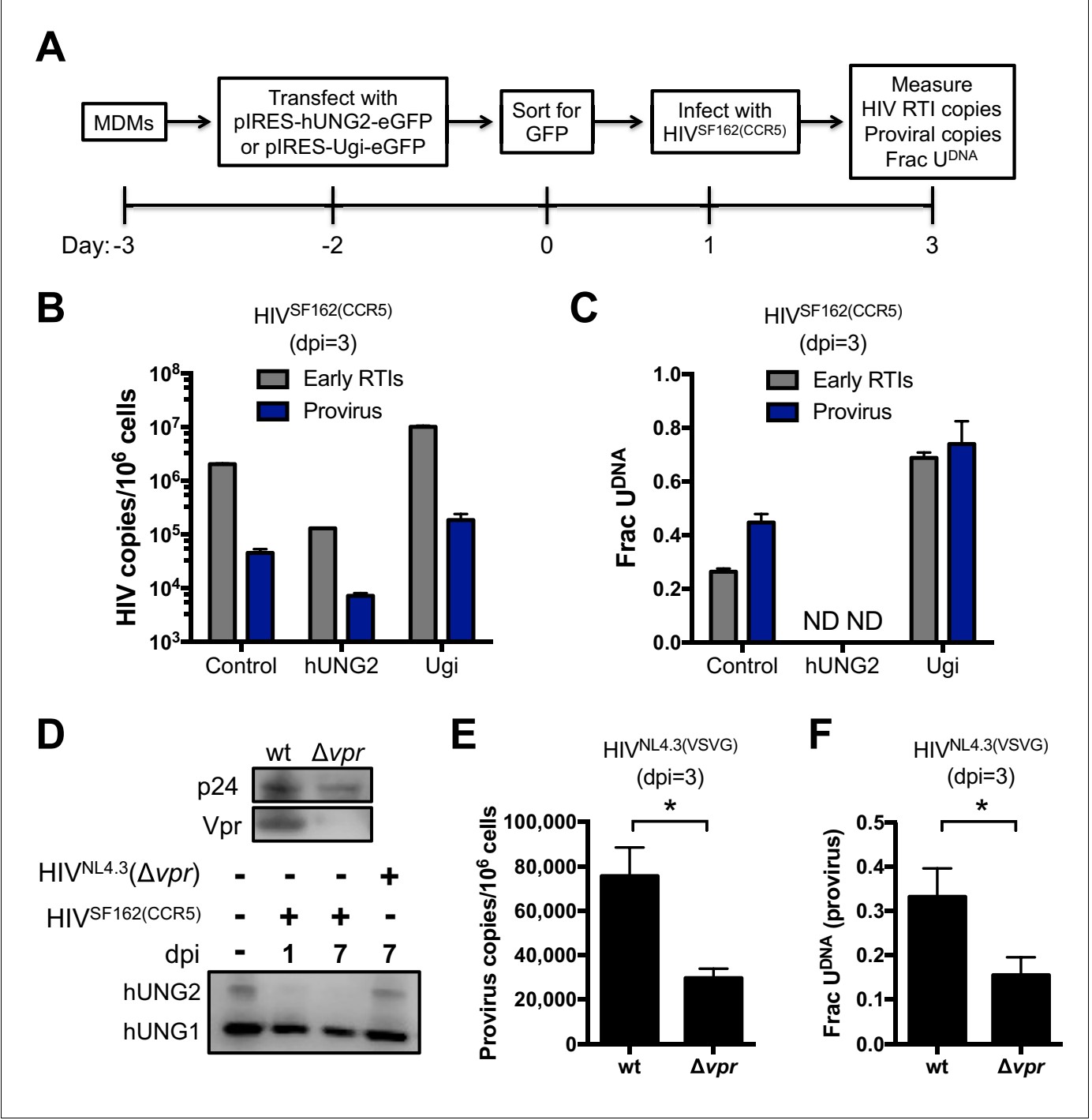

**Figure 4.** hUNG2 uracil excision activity are required for efficient pre-integration restriction. Effect of transient overexpression of nuclear hUNG2 and Ugi on copies of viral early reverse transcription intermediates (RTIs) and proviruses. (**A**) Experimental scheme for hUNG2 and Ugi overexpression in MDMs. (**B**) Copies were determined by qPCR using specific primers for early RTIs and provirus. (**C**) Fraction of viral RTIs and provirus that contained detectable uracil under conditions of hUNG2 depletion (Ugi) and overexpression (hUNG2). ND; Not detectable. (**D**) Western blot analysis of the *vpr* deletion virus HIV[NL4.3](Δvpr) revealed no detectable Vpr. Western blot analysis to detect nuclear (hUNG2) and mitochondrial (hUNG1) hUNG isoforms detected oscillation in the hUNG2 levels over the course of a 7-day multi-round infection. At day 7, the HIV[NL4.3](Δvpr) shows abundant hUNG levels while hUNG2 is depleted in the wild-type infection. (**E**) Measurements of provirus copy number in a mixed population of GFP[−] and GFP[+] MDMs infected with HIV[NL4.3(VSVG)] (wt) and HIV[NL4.3](Δvpr). (**F**) Ex-*ALU-gag* nested qPCR measurements of the fraction of uraciated proviral copies in a mixed population of GFP[−] and GFP[+] MDMs infected with HIV[NL4.3(VSVG)] (wt) and HIV[NL4.3](Δvpr) at 3 days post-infection. Level of statistical significance (*$p<0.05$). Number of experimental replicates (n = 4–5) and errors are reported as mean ± SD.

*Figure 4 continued on next page*

*Figure 4 continued*

The following source data is available for figure 4:

**Source data 1.** Ex-*ALU-gag* qPCR for measurement of reverse transcription intermediates (RTIs), provirus and uracil content in MDMs transfected with plasmids encoding hUNG2, Ugi or empty control (*Figure 4B,C*).

sorted by FACS 48 hr post-transfection and the GFP⁺ cells (indicating successful plasmid transfection; not expression of HIV-encoded GFP) were allowed to adhere for 24 hr and then spin-infected with HIV$^{SF162(CCR5)}$ virus. Cells were harvested 3 days post-infection and total DNA was extracted for qPCR analyses.

For the cells that were transfected with the Ugi expression plasmid, the copy number of early reverse transcription (RTI) products and proviral DNA was increased about five-fold compared to transfection with the empty control plasmid (*Figure 4B*). The Ugi-expressing cells had dramatically increased uracil content, consistent with efficient knockdown of hUNG2 activity and a prominent role for hUNG2 in uracil excision (*Figure 4C*). In contrast, the cells transfected with the hUNG2 expression plasmid showed a 10-fold reduction in RTIs and proviral DNA relative to control (*Figure 4B*), and we were unable to detect any uracil in viral DNA using Ex-qPCR (*Figure 4C*). These results closely match our previous findings where high expression levels of hUNG2 served to effectively destroy uracilated viral DNA (*Weil et al., 2013*).

In a complementary approach to establish a role for hUNG2 in uracil excision, we infected the mixed population of MDMs with HIV$^{NL4.3(VSVG)}$, HIV$^{SF162(CCR5)}$ or HIV$^{NL4.3}$(Δ*vpr*) viruses (*Figure 4D–F*). Infections with HIV$^{SF162(CCR5)}$ and HIV$^{NL4.3}$(Δ*vpr*) showed that the hUNG2 nuclear isoform (but not the mitochondrial hUNG1 isoform) selectively disappeared between 1 and 7 days post-infection with HIV$^{SF162}$(*vpr*⁺), but not with HIV$^{NL4.3}$(Δ*vpr*) (*Figure 4D*) (*Eldin et al., 2014*). As expected, infection with HIV$^{NL4.3}$(Δ*vpr*) reduced the overall proviral copies present in the mixed population of GFP⁺ and GFP⁻ MDMs (*Figure 4E*), which is a non-specific effect that can be attributed to any of the known pro-infective functions of vpr (*Eldin et al., 2014*). In the absence of vpr, two-fold fewer proviruses contained uracil, which is a uracilation-specific effect that can be attributed specifically to vpr-induced depletion of hUNG2 activity in GFP⁻ MDMs (*Figure 4F*) (*Grogan et al., 2011*). The larger effects observed above with Ugi and hUNG2 overexpression are expected because the hUNG2 level is manipulated before viral infection.

## Genomic RNAs are produced from uracilated proviruses

We investigated whether the presence of uracil in proviral DNA affected the efficiency of producing viral RNA genome copies. In these experiments, we use reverse transcriptase quantitative PCR (RT-qPCR) to measure the copy number of extracellular viral genomic RNAs (EVRs) present in supernatants from infected GFP⁺ and negative MDMs that were cultured in standard growth media or media supplemented with IFNγ or IL-4 (*Figure 5A*). These two cytokines were of interest because they stimulate macrophage differentiation into classical pro-inflammatory (M1) and anti-inflammatory (M2) polarization states, respectively, and could modulate the outcome of infection (*Xue et al., 2014*). The EVR copies present in the supernatant of GFP⁻ MDMs at three days post-cytokine stimulation was about 100-fold lower than the GFP⁺ population, independent of cytokine stimulation. The lower copies of genomic RNA produced from the uracilated proviruses in GFP⁻ MDMs is comparable to the reduced proviral copy number in this population. This result indicates that uracilation does not significantly alter the output of viral RNA genomes when normalized for proviral DNA levels. We also found that inflammatory cytokine IFNγ stimulation reduced the EVR copy number in both MDM populations by almost 10-fold (*Figure 5B*), which is therefore not correlated with the uracilation phenotype.

## Extracellular viral RNAs (EVRs) have more mutations than uracilated proviral DNA

Limiting dilution clonal sequencing was performed on EVRs produced from sorted GFP⁻ and GFP⁺ MDMs at 3 days post-stimulation (*Laird et al., 2013*; *Ho et al., 2013*), and revealed surprising

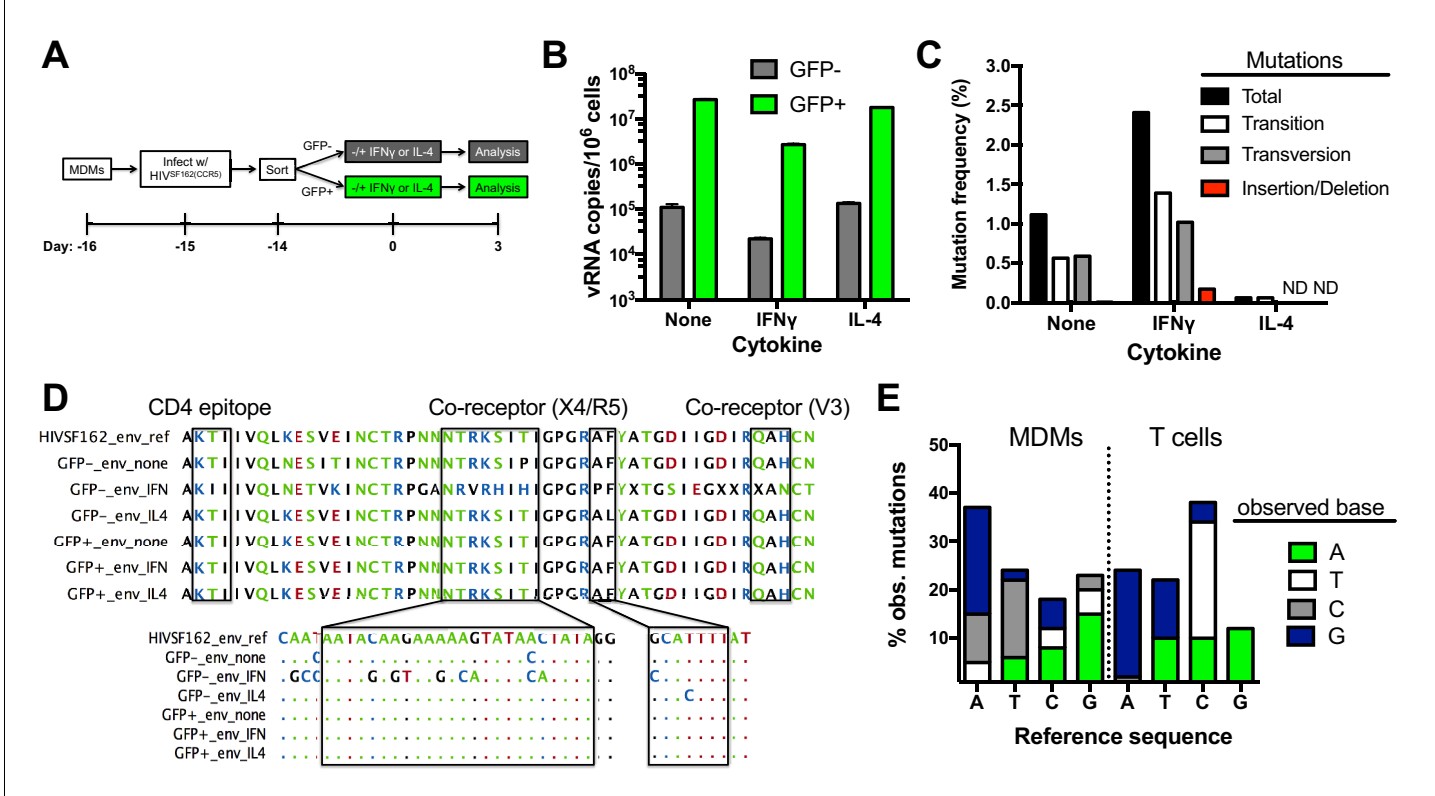

**Figure 5.** Impact of uracilation and cytokines on viral DNA and RNA sequences. (A) Experimental protocol. (B) Quantitative reverse transcriptase-PCR (qRT-PCR) was used to determine the copy numbers of extracellular viral RNA in cell supernatants from sorted GFP+ (green bars) and GFP− (white bars) MDMs with and without cytokine stimulation. Number of experimental replicates (n = 4) and errors are reported as mean ± SD. (C) Summary of mutations from limiting–dilution clonal sequencing of extracellular viral RNAs obtained from infected GFP− MDMs under different growth conditions. The mutation frequencies (point mutations per total nucleotides sequenced) were obtained from a ~500 bp amplicon of the *env* gene. ND; not detected. (D) Representative *env* sequences of extracellular viral RNAs produced by GFP− sorted MDMs. The top sequence is for the HIV^SF162(CCR5) strain used to infect the MDMs (HIVSF162_*env*_ref). Boxed regions at the protein and nucleotide sequence level show the mutation spectrum within the CD4-associated and co-receptor binding sites. (E) Breakdown of point mutations for viral RNA produced from infected MDMs and proviral DNA from infected T cells (see text).

The following source data is available for figure 5:

**Source data 1.** qPCR measurement of gag copies (*Figure 6B*), provirus copies (*Figure 6C*) and uracil content (*Figure 6D*) in GFP sorted and cytokine-treated MDMs.

differences in viral single-point mutation frequencies for these populations (*Figure 5C*). Regardless of cytokine stimulation, the GFP+ population showed no detectable mutations in the roughly 12,000 total bases of the V3/V4 variable region of *env* that were sequenced. In contrast, the GFP− population showed a 1.2% basal mutation frequency, which responded differently to IFNγ and IL-4 stimulation: IFNγ stimulation increased the frequency by two-fold and IL-4 decreased it significantly (*Figure 5C*). These results suggest that the cytokine environment of an in vivo infected macrophage could influence the evolution of uracilated proviruses. In contrast to the mutated *env* sequences, the LTR region of EVRs produced from the GFP− cells was devoid of detectable mutations after sequencing 7392 total bases (*Supplementary file 1*), which may reflect selection for expression of transcription competent LTR sequences. The absence of mutations in the *env* and LTR regions of EVRs isolated from the GFP+ MDM cultures, as well as the LTR region of EVRs obtained from the GFP− culture supernatants, virtually eliminates the possibility that the observed mutations in *env* arise from PCR or sequencing errors.

Targeted sequencing of the *env* amplicon in uracilated viral DNA isolated from GFP− MDMs at day 14 post-infection in the absence of cytokine stimulation revealed a <0.007% mutation frequency

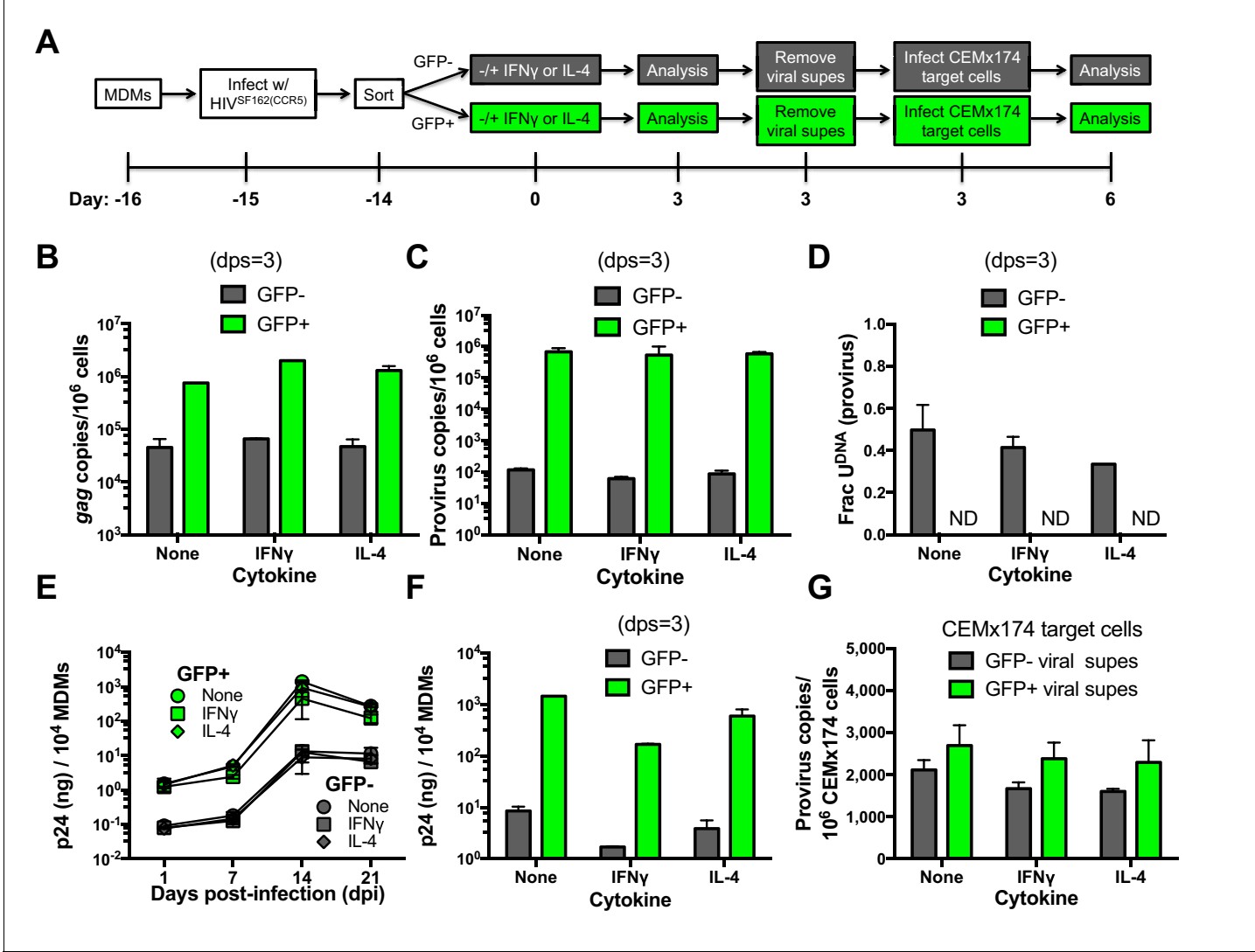

**Figure 6.** Effects of cytokine stimulation on viral transmission in GFP sorted MDM populations. (**A**) Experimental approach. (**B**) MDMs infected with HIV[SF162(CCR5)] were sorted into GFP+ (green bars) and GFP− (white bars) populations and analyzed with respect to *gag* copy number (dps; days post-stimulation). (**C**) Provirus copy numbers for each MDM population were measured using *ALU-gag* nested qPCR at 3 dps. (**D**) The fraction of proviral DNA copies containing uracil was measured using Ex-*ALU-gag* nested qPCR. ND; Not Detected. (**E**) Viral growth kinetics of sorted MDM populations. (**F**) Levels of virus in culture supernatants of GFP+ and GFP− MDM producer cells were measured using p24 ELISA. (**G**) Viral supernatants from each MDM treatment were normalized to p24 levels and used to infect naïve CEMx174 target cells. Provirus copies were measured using *ALU-gag* qPCR. The results are normalized to one million target cells. In all cases, the number of experimental replicates (n = 4–5) and errors are reported as mean ± SD.

The following source data is available for figure 6:

**Source data 1.** qPCR measurement of gag copies (*Figure 6B*), provirus copies (*Figure 6C*) and uracil content (*Figure 6D*) in GFP sorted and cytokine treated MDMs.

in the V3/V4 variable region, which is significantly lower than the 1.2% frequency in EVRs produced from the same cell culture (most viral DNA is integrated at day fourteen). This result suggests that the EVR mutations arise from transcriptional errors by RNA polymerase II (RNAP II). Such mutations may derive from encounter of the polymerase with either uracils or repair intermediates resulting from uracil excision (see Discussion). Illumina next-generation sequencing data covering the entire viral DNA genome (obtained from total HIV[NL4.3] viral DNA isolated from a mixed population of

MDMs at day 7 post-infection in the absence of cytokines) confirmed the low mutation frequency determined from targeted sequencing of the *env* V3/V4 variable region (see Data Access information).

The frequencies of single-base transition, transversion, and insertion/deletion mutations within the *env* region of EVRs produced from unstimulated and IFNγ or IL-4- stimulated GFP⁻ cells were calculated from the sequencing data (*Figure 5C*). The RNA (+) strand mutations were primarily single base changes mostly comprised of A→G, A→C, and T→C transitions and transversions (*Figure 5D,E*), but also a few short 2–3 nucleotide insertions or deletions. A detailed analysis of the sequence dependence of the observed G→A (+) strand mutations using *Hypermut* 2.0 (*Rose and Korber, 2000*) indicated that the sequences and frequencies were not consistent with A3G, A3F or A3A deaminase activity on the viral (-) strand cDNA (*Yu et al., 2004*).

To explore whether the mutation signature of uracilated viral DNA products and viral genomic RNA sequences derived from infected MDMs were distinct from those obtained from T cells, we infected activated CD4+ T cells with HIV^BAL(CCR5) virions produced from donor cells expressing endogenous levels of A3G (Materials and methods). The infection was kept to a single round by introducing the entry inhibitor enfuvirtide after the initial infection (*Hildinger et al., 2001*), and the V3/V4 *env* region of 14 proviruses were sequenced. The proviral sequences derived from infected T cells had a much higher mutation frequency (~1%) than uracilated viral DNA in MDMs (<0.007%) (*Supplementary file 2*). In addition, the mutation spectrum of proviral DNA isolated from T cells differed from the EVRs produced from MDMs (*Figure 5E*). The prominent proviral mutations found in T cells were elevated C→T transitions and T→G transversions, while the viral RNA sequences derived from MDMs were distinguished by elevated A→C transversions and T→C transitions (*Figure 5E*). We note that previous studies have established that host RNAP II contributes little to HIV sequence evolution in T cell infections and that errors during reverse transcription are the predominant diversification mechanism (*Menendez-Arias, 2009*). In contrast, for in vitro infection of MDMs a majority of the mutations occur during transcription.

## Uracilated proviruses produce replication competent HIV

To measure the levels of functional progeny viruses that emerged from infected MDMs in the absence and presence of cytokine stimulation, we followed the approach outlined in *Figure 6A*. MDMs infected with HIV^SF162(CCR5) were first sorted based on GFP fluorescence at one day post-infection. The sorted GFP positive and negative cells were divided into three cultures that were stimulated with IFNγ, IL-4, or no treatment. After 3 days, the MDM producer cultures were analyzed with respect to (i) *gag* DNA copy number as a surrogate for total viral DNA present (*Figure 6B*), (ii) integrated proviral DNA copy number (*Figure 6C*), (iii) the fraction of proviruses containing uracil (*Figure 6D*), and (iv) p24 levels in culture supernatants (*Figure 6E*). The virus-containing MDM culture supernatants were removed at three days post-stimulation (dps), normalized to p24 and added to cultures of CEMx174 target cells. After 3 days, the proviral copy number per 10⁶ target cells was measured (*Figure 6F*).

Independent of stimulation, the GFP⁻ MDM producer cells showed lower copy numbers of *gag* and proviral DNA (~20 and 10⁴-fold) and p24 levels (100-fold), as compared to GFP⁺-MDMs (*Cassetta et al., 2013*). IFNγ or IL-4 stimulation had little effect on *gag* and proviral DNA copy numbers in both MDM producer populations (*Figure 6B,C*), but IL-4 induced a modest decrease in the fraction of proviruses that contained uracil at 3 days post-stimulation (*Figure 6D*). For both MDM types, the addition of either cytokine resulted in a reduction in p24 production, with IFNγ showing the largest effect relative to no stimulation (~10-fold decrease) (*Figure 6E,F*). Despite the very restrictive environment of GFP⁻ MDM producer cells prior to integration (*Figure 3D*), the harvested viruses derived from uracilated proviruses were highly competent for infection of CEMx174 target cells (*Figure 5G*).

## Uracils are present in HIV-1 DNA from peripheral blood monocytes of virally suppressed donors

To test whether HIV-1 DNA isolated from infected individuals contained detectable levels of uracil, we purified resting CD4⁺ T cells and monocytes from bulk peripheral blood mononuclear cells (PBMCs) of six participants suppressed on antiretroviral therapy (ART). Highly purified populations of

resting CD4+ T (>95%) cells and monocytes (>98%) were obtained and quantified by flow cytometry (*Figure 7—figure supplement 1A–C*). Both ddPCR and Ex-ddPCR were used to quantify HIV DNA and measure the fraction of the viral *pol* amplicons that contained uracil (*Figure 7A,B*) *Strain, 2013*. HIV DNA levels in resting CD4+ T cells were generally higher and more variable than those measured in monocytes (geometric mean values of 526 copies/$10^6$ T cells and 47 copies/$10^6$ monocytes) (*Figure 7A*) (*Wang et al., 2013*; *Eriksson et al., 2013*). Consistent with in vitro measurements (*Figure 1B*), we found no detectable uracil (Frac U) in HIV DNA isolated from resting CD4+ T cells, but five of the six donor monocyte samples contained detectable uracils. The fraction of uracil-positive *pol* amplicons was in the range 20 to 80% for the five patient samples (*Figure 7B*). The observation that viral DNA uracilation is specific to monocytes and macrophages provides an unambiguous marker of its origins and excludes contaminating T cell DNA as a possible source.

We also obtained access to cryopreserved bronchial alveolar lavage (BAL) and PBMC isolates from a single HIV-1 infected donor that were collected prior to initiation of ART and six months after ART therapy and performed a similar analysis (*Figure 7B*). Viral DNA from highly purified monocytes was tested, as well as bronchial alveolar lavage (BAL) samples that consisted of >95% alveolar macrophages (AMs). Using the Ex-*ALU-gag* PCR method for detecting proviral DNA, we found that ~40% (monocytes) and 80% (alveolar macrophages) of the *ALU-gag* amplicons were positive for uracils. Similar levels of uracils were detected in pre- and post-ART BAL isolates (*Figure 7B*). Although a stable reservoir for HIV is primarily located in resting CD4+ T cells, these results suggest that HIV DNA residing in monocytes/alveolar macrophages comprises another potential source for viral rebound after discontinued therapy, consistent with other reports (*Cribbs et al., 2015*).

## Discussion

### Host cell DNA repair, HIV restriction, persistence and mutagenesis

Previous literature contains diverse and sometimes conflicting data on the role of dUTP and host cell uracil DNA glycosylase in HIV infection [reviewed in *Weil et al. (2013)*]. Our key finding that MDMs consist of two distinct populations with respect to the uracilation phenotype likely explains why this potent restriction pathway has remained largely elusive.

The established elements of this pathway are depicted in *Figure 7C*. Upon entry into the macrophage, reverse transcriptase (RT) encounters a nucleotide pool environment that favors the incorporation of dUTP into HIV DNA products predominantly in the form of U/A base pairs. The high ratio of dUTP/TTP is maintained, at least in part, by the low levels of dUTPase expression combined with the high expression levels of SAMHD1 in macrophages. Uracilation of the viral DNA can proceed to high levels in the cytoplasmic compartment because the UBER machinery is sequestered in the nucleus. However, when the heavily uracilated HIV DNA enters the nucleus it is attacked and fragmented by nuclear UBER enzymes. Fragmentation is initiated by hUNG2 uracil excision, and is likely followed by abasic site excision by human abasic site endonuclease (APE1 or APE2) (*Figure 1—figure supplement 1*) (*Schrader et al., 2009*; *Stavnezer et al., 2014*). We estimate, based on comparing the measured levels of early and proviral DNA levels in the GFP⁻ MDMs (*Figure 3*), that ~99% of the viral DNA is destroyed before the integration step and further destruction at the proviral stage is observed over the next 14 days (*Figure 3D*). This pathway differs from original models of APO-BEC3G-mediated virus restriction in that uracils are introduced on both DNA strands and they are not inherently mutagenic (*Priet et al., 2006*).

We speculate that lethal mutagenesis, functional genome variation, or faithful repair could result from host enzyme processing of uracilated proviruses (*Figure 7C*). In non-dividing MDMs, we found that uracilated proviruses contained few mutations yet gave rise to mutated viral genomic RNAs (*Figure 5D*). Although it is tempting to attribute these mutations to RNAP II transcriptional errors as it encounters uracil on the template DNA strand, RNAP II shows high-fidelity incorporation of A opposite to U during in vitro transcription *Menendez-Arias, 2009*. Moreover, many aspects of the mutation spectrum in *Figure 5E* are not reconcilable with RNAP II incorporating incorrect bases opposite a template U (although the frequent A→G transitions may be explained by the potential of U to pair with A or G) (*Kuraoka, 2002*). The actual error mechanism is likely to be much more complex given that transcription coupled repair is expected to generate abasic sites, strand breaks, and even gaps when densely spaced uracils are excised. In this regard, RNAP II is known to stall at

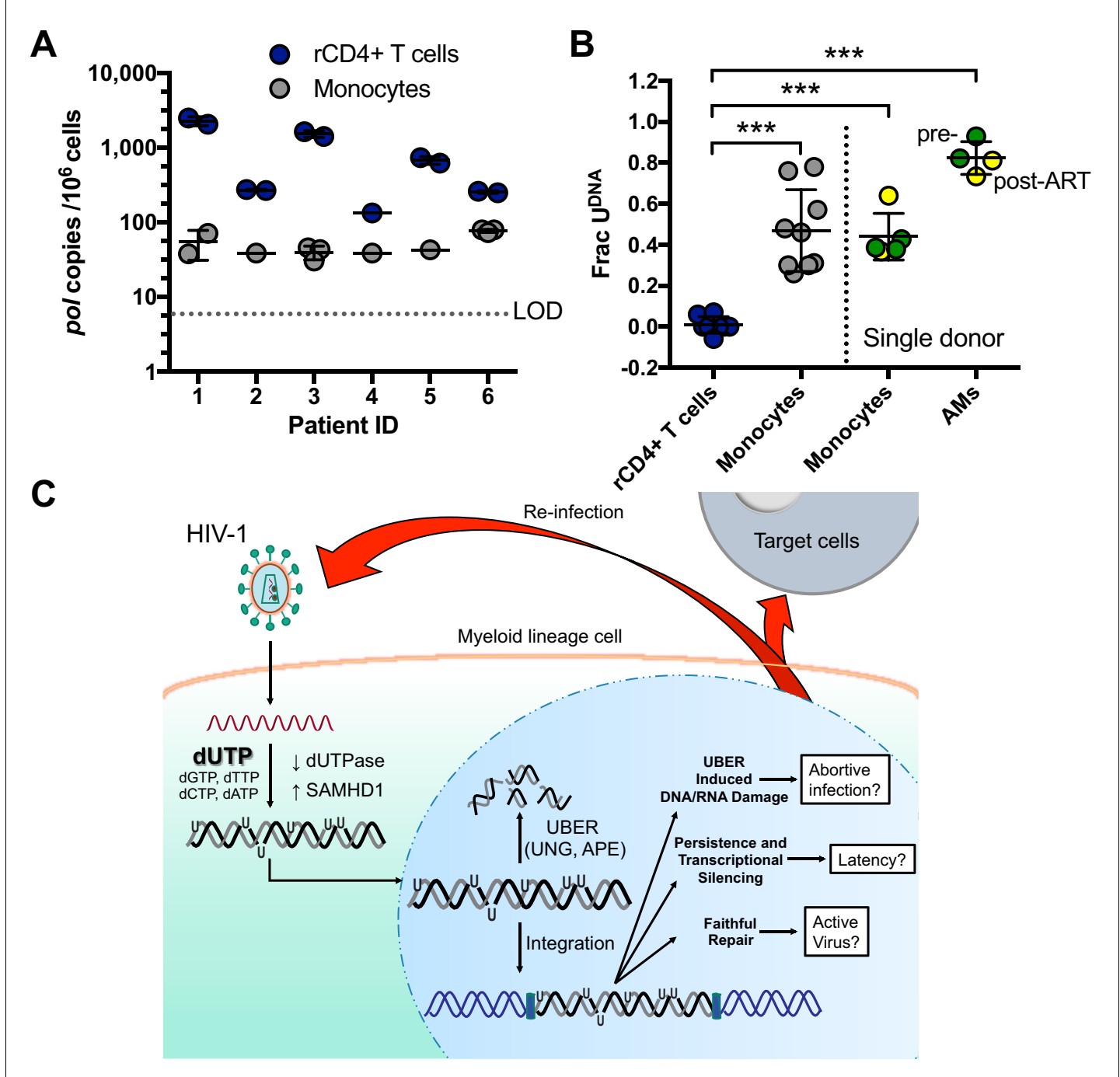

**Figure 7.** Peripheral blood monocytes and alveolar macrophages contain high levels of uracil in HIV DNA. (**A**) Resting CD4+ (rCD4+) T cells and monocytes were purified by negative bead selection from bulk PBMCs obtained from six ART-suppressed individuals. Total HIV$^{pol}$ DNA was quantified using Ex-ddPCR. Copy numbers were normalized to the human *RPP30* gene to give an estimate of HIV copies/10$^6$ cells. Limit of detection (LOD) = 5 copies/10$^6$ cells. (**B**) The uracilated fraction of HIV *pol* DNA copies derived from monocytes and rCD4+ T cells was measured using Ex-ddPCR. Total genomic DNA was also isolated from matched, cryopreserved PBMCs and bronchial alveolar macrophages (AM) obtained from a single donor both pre-ART (green circle) and post-ART (yellow circle), (***p<0.001). Donor 4 had no detectable uracil (*pol*) and was excluded from this plot. (**C**) Infection of MDMs by HIV-1 and possible fates of uracilated viral DNA products (see text). Level of statistical significance (***p<0.001). Each experimental replicate is shown. Number of experimental replicates (n = 1–3) and errors are reported as mean ± SD.

The following figure supplement is available for figure 7:

**Figure supplement 1.** Purity and detection of HIV DNA in isolated cell populations from ART suppressed individuals.

isolated abasic sites during in vitro transcription and could also misincorporate various ribonucleotides at such sites in vivo (*Yu et al., 2003*).

The observation that proviral DNA is relatively mutation free, while the RNA is not, strongly suggests that repair intermediates generated during transcription by the host UBER pathway are most often repaired to restore the proviral coding sequence. Faithful repair could involve the reincorporation of U opposite to A due to the perturbed nucleotide pools, or the introduction of T. The observation that RNA mutations were increased upon stimulation with the inflammatory cytokine IFNγ, and nearly absent in the presence of the anti-inflammatory cytokine IL-4, indicates that the transcriptional programs induced by these cytokines are highly relevant to the outcome; a further understanding of this complex problem is of considerable interest. Although errors introduced by IFNγ stimulation may frequently lead to non-functional viral genomes, they also allow the viral DNA sequence to evolve rapidly in the absence of reverse transcription. Replication-competent viruses that emerge would then undergo fitness selection in the host. These potential outcomes make UBER a double-edged sword with respect to propagation of viral infection.

## Uracil as an epigenetic factor in macrophages

Uracil has never been considered an epigenetic base like 5-substituted cytosines because of its low abundance in genomes and because the detection of U/A base pairs is not possible with standard sequencing methods. Nevertheless, the densely incorporated uracils detected here have the potential to alter DNA structure and dynamics (*Ye et al., 2012*), which could affect site recognition by transcription factors or other DNA binding proteins. There is a growing body of evidence indicating that uracil can exert significant effects on protein binding to DNA. These findings include (i) our previous report that multiple uracils silenced transcription from both HIV-1 LTR and CMV promoters in a cell line (*Weil et al., 2013*), (ii) uracils within the origin of replication in HSV-1 perturb the binding of HSV-1 origin binding protein (*Focher et al., 1992*), (iii) U/A pairs disrupt AP-1 transcription factor DNA binding (*Rogstad et al., 2002*), (iv) singly uracilated DNA disrupts RNase H splicing specificity during reverse transcription *Klarmann, 2003*, (v) U/A pairs perturb maintenance of telomere length in B cells by disruption of sheltrin binding (*Vallabhaneni et al., 2015*), and (vi) one or two U/A base pairs within the specific cleavage site of some restriction enzymes prevents DNA strand cleavage (*Roberts et al., 2015*). In addition, the abasic site product of uracil excision is known to exert a large negative effect on transcription (*Luhnsdorf et al., 2014*) as would any mutations in transcription factor recognition sequences arising from error prone repair of excised uracils (*Shah et al., 2015*; *Emiliani, 1998*). Combined, these potential effects of U/A pairs could profoundly silence HIV gene expression.

## Uracilation as a biomarker for ongoing infection

Historically, macrophages have been controversial targets for HIV infection (*Sattentau and Stevenson, 2016*). Macrophages have unique cellular phenotypes that are conducive for a significant impact on infection because they persist for months after initial infection (*Gerngross and Fischer, 2015*), continue to produce infectious virus particles for their entire lifetime (*Crowe and Sonza, 2000*), and provide a drug tolerant environment for HIV propagation (*Gavegnano and Schinazi, 2009*; *Gavegnano et al., 2013*). Moreover, their residence, within tissues that have poor drug penetrance (lung, brain, gut) (*Cribbs et al., 2015*), has led to an increased interest in their potential role in the establishment and persistence of HIV infection.

The detection of abundant viral uracils in monocytes isolated from HIV-1 infected individuals who showed full viral suppression from ART, in vivo The fairly short several day half-life of blood monocytes suggests that uracilated viral DNA detected in these cells arose from a recent infection, perhaps by passage of the monocyte through a tissue reservoir containing virus producing cells (*Wang et al., 2013*). Such an infection pathway could involve a canonical CCR5-dependent viral entry mechanism or a recently described pathway involving phagocytosis of infected T cells by macrophages (*Baxter, 2014*). However, the high efficiency of ART and the low levels of infected T cells using current treatment regimes make the phagocytosis pathway for monocyte infection seem less plausible for our aviremic patient samples. Finally, the requirement for dUTP incorporation during reverse transcription virtually excludes the possibility that the infected monocytes containing uracilated DNA arose from latently infected myeloid stem cells, because monocyte differentiation does

not involve reverse transcription and genomic DNA replication does not involve accumulation of detectable uracils (*Figure 7—figure supplement 1I*).

A further ramification of these data is that some uracilated proviruses may persist for the entire lifetime of a macrophage due to protection within highly compacted chromatin, which is highly resistant to UBER (*Ye et al., 2012*). Under such circumstances, the potential time scale for uracil persistence could be years based on the findings that replication-competent viruses have been detected in brain microglial cells of patients even after years of viral suppression using ART (*Rappaport and Volsky, 2015*; *Robillard et al., 2014*). If U/A pairs are found to down regulate viral gene expression through chromatin stabilization or by weakening the binding of transcription factors, some proviruses could remain transcriptionally silenced for years and then become activated when appropriate stimulatory conditions are present.

## Materials and methods

### Study subjects

This study was approved by the Johns Hopkins Institutional Review Board and written informed consent was provided by both HIV-infected and healthy individuals (IRB00038590). Peripheral blood was obtained from healthy volunteers and ART suppressed patients with viral loads <20 copies HIV-1 RNA/mL. Monocytes and resting CD4+ T cells were obtained from donor PBMCs as described in *Isolation of primary cells from donor PBMCs* and the purity of the cell types was determined by flow cytometry.

### Viruses

Vesicular stomatitis virus G protein (VSV-G) pseudo-typed HIV-1 virions (HIV$^{NL4.3(VSVG)}$) were generated as previously described (*Weil et al., 2013*) by co-transfection of HEK 293T (RRID:CVCL_0063) cells with pNL4-3-ΔE-eGFP and pVSV-G. Virions containing the CCR5 co-receptor (HIV$^{SF162(CCR5)}$) were obtained by transfection of HEK 293T cells with pSF162R3-GFP-Nef$^+$. HIV$^{BAL(CCR5)}$ virus was obtained from NIH AIDS Research and Reference Reagent program (cat #510). Viral supernatants were collected 48 hr after transfection, DNaseI-treated to remove any plasmid carryover and virions were purified over a 20% (wt/vol) sucrose cushion. HIV$^{NL4.3}$(Δ*vpr*) mutant virus construct was generated by site-directed mutagenesis of pNL4-3-ΔE-eGFP. Primers used for site-directed mutagenesis can be found in *Supplementary file 3*.

### Isolation of primary cells from donor PBMCs

Peripheral blood mononuclear cells (PBMCs) were isolated from HIV-1 positive or negative donors using density centrifugation on a Ficoll-Hypaque gradient. Monocytes were isolated from the PBMC population using the Pan monocyte Isolation Kit (Miltenyi Biotec). Monocytes were differentiated into macrophages over 7 days using MDM-20 media containing RPMI 1640, 20% (vol/vol) autologous plasma, 10 ng/mL GM-CSF or M-CSF (BD Biosciences, San Jose, CA), 1 × HEPES, and 1 × Glutamine (Gibco, Gaithersburg, MD). Cultured MDMs were maintained in media containing RPMI 1640 + 10% (vol/vol) dialyzed FBS, 1% Pen/Strep. CD4+ T cells were isolated from PBMC population using the CD4+ isolation kit (CD4+ T cell Isolation kit II, Miltenyi Biotec, San Diego, CA). CD4+ cells were activated with 0.5 µg/mL phytohemagglutinin (PHA) for 3 days in IL-2 containing media. Resting CD4+ cells were further enriched from the bulk CD4+ cell population by negative selection against biotinylated activation markers; CD25, CD69 and HLA-DR (Miltenyi Biotec).

### Purity of isolated cell populations

The purity of monocytes, activated and resting CD4+ cells was determined by flow cytometry with fluorescent labeling of anti-human; Fluorescein isothiocyanate (FITC)- and allophycocyanin (APC)-conjugated mAbs CD4 (RRID:AB_314074), CD25 (RRID:AB_2280228), CD69 (RRID:AB_492844) and HLA-DR (RRID:AB_10839413). Isotype-matched control mAbs (BD Biosciences) were used for gating and quantification. Monocyte and MDM populations were stained using brilliant violet 421-conjugated (BV421) anti-CD14 (RRID:AB_10899407), CD16-APC (RRID:AB_2616904) and CD3-FITC (RRID: AB_2616618). The purity of these cells were analyzed by flow cytometry using a FACS Canto II (BD Biosciences) and FlowJo software (Treestar). Using this method, we obtained monocytes with a

purity of >99.9% with undetectable T cell contamination. Cells were then pelleted and used for protein extraction, dNTP extraction, or infection as described above; for cell infections, cells were analyzed for GFP expression by FACS analysis.

## Fluorescence activated cell sorting

In preparation for cell sorting studies, MDMs infected with GFP-reporter virus (HIV[NL4.3(VSVG)] or HIV[SF162(CCR5)]) were washed twice with Hank's balanced salt solution (HBSS) and then detached by incubating with Accutase (Cell Technologies) for 30 min at 37°C. MDMs were resuspended at $5 \times 10^6$ cells/mL in $1 \times$ HBSS (pH 7.2), 5 mM EDTA and 0.5% BSA. Prior to sorting, cell suspensions were passed through a 35 mm nylon mesh (BD Biosciences) and propidium iodide added to gate out dead cells. Sorting was performed on a MoFlo Sorter using a 100 μm nozzle and the sort purify 1 mode.

## Transient overexpression of nuclear hUNG2 and Ugi

pIRES-Ugi-eGFP and pIRES-hUNG2-eGFP reporter plasmids were constructed by replacing the neoR cassette for eGFP using SmaI and XbaI cloning sites of pIRESneo3 (Clonetech, Mountainview, CA). The empty pIRES-eGFP plasmid was used as a transfection control (shown). Primary macrophages derived from blood monocytes were cultured in the presence of M-CSF for 7 days prior to transfection using jetPEI-Macrophage in vitro DNA transfection reagent (Polyplus). Cells expressing GFP were sorted by FACS 48 hr post-transfection. GFP+ cells (indicating the presence of hUNG2 or its inhibitor protein) were allowed to adhere for 24 hr and then spin-infected with HIV[SF162(CCR5)] virus. Cells were harvested 3 days post-infection, and DNA was extracted using QIamp DNA kit (Qiagen, Valencia, CA) for PCR and Ex-PCR analyses. Due to the low efficiency of transfection using MDM target cells, it was not possible to obtain immunoblots for Ugi or hUNG2 expression levels in these samples. However, the specific and opposite effects of the Ugi and hUNG2 transfection, the absence of an effect upon transfection with pIRES-eGFP plasmid, and the alternative approach of deleting *vpr* are strongly supportive of our interpretations.

## Ex vivo enzyme activity assays

Protein lysates were obtained using CelLytic M (Sigma, St. Louis, MO) reagent according to the manufacturer's instructions. SAMHD1 activity was determined as previously described with several modifications (*Hansen et al., 2014*). Protein lysates (5 μg) were incubated in buffer containing; 5 mM [8-³H] dGTP (activator/substrate) (Moravek Biochemicals, Brea, CA), 5 mM ATP, and 10 mM paranitrophenyl phosphate (p-NPP) in a total volume of 50 μL. ATP was included to inhibit nonspecific phosphatases and p-NPP was added to prevent the degradation of dUTP by alkaline phosphatase (*Williams and Parris, 1987*). The small molecule inhibitor (pppCH₂dU) of SAMHD1 was used to determine specificity of the assay. Reactions were incubated at 37°C at varying time points, at which 2-μL samples were removed and quenched by spotting onto a C18-reversed phase thin layer chromatography (TLC) plate. The TLC plate was developed in 50 mM KH₂PO₄ (pH 4.0) to separate substrate [dGTP ($R_f$ = 0.80)] from products [dG ($R_f$ = 0.20)]. Plates were exposed to a tritium-sensitive screen for 5 hr and then scanned on a Typhoon phosphoimager (GE Healthcare, Pittsburgh, PA) and the counts present in the substrate and product were quantified using the program Quantity One (Bio-Rad, Hercules, CA).

dUTPase activity was assessed as previously described (*Weil et al., 2013*). Briefly, protein lysates were incubated with [5-³H] dUTP (Moravek Biochemicals, Brea, CA) and the fraction of substrate hydrolyzed to dUMP product was monitored. Reactions were incubated at 37°C for 1 hr, after which 2-μL samples was spotted onto a PEI-cellulose TLC plate. The TLC plates were developed in 0.5 M LiCl to separate substrate [dUTP ($R_f$ = 0.1)] from the product [dUMP ($R_f$ = 0.6)]. Plates were exposed to a tritium-sensitive screen for 5 hr and scanned on a Typhoon phosphoimager (GE Healthcare).

A molecular beacon hairpin reporter assay was used to determine endogenous UNG activity in crude cell extracts (*Weil et al., 2013*). Reactions were performed using 5 μg of protein lysate and 50 nM S-pin-18 in 10 mM Tris-HCl (pH 7.1), 100 mM NaCl, 1 mM EDTA, and 0.2% Triton X-100. Supplementing the reaction with the potent Uracil DNA glycosylase inhibitor (Ugi) showed no measurable activity. Additionally, a molecular beacon assay was used to monitor Apurinic/Apyrimidinic endonuclease (APE) activity (*Seiple et al., 2008*). Two FAM and dabcyl quenched DNA duplex substrates

were used. A specific substrate (SS) containing a single abasic site ($\Phi$) and a non-specific substrate (NS) with the abasic site replaced with the canonical DNA base Thymine (T). The sequence of the specific molecular beacon substrate; SS, 5'-FAM-GAGAA$\Phi$ATAGTCGCG3' and 5'-CGCGACTATG TTCTC-dabsyl-3' (where $\Phi$ is a tetrahydrofuran abasic site analog). Independent reactions were performed using 5 µg of protein lysate and 50 nM in 10 mM Tris-HCl (pH 7.1), 100 mM NaCl, 1 mM EDTA, and 0.2% Triton X-100. To determine APE activity, the difference in the initial rate of the specific substrate from the non-specific rate was measured. Rates were measured on a Fluoro-Max 3 (Horiba Jobin Yvon, Edison, NJ).

## dNTP extraction, rNTP removal and LC-MS quantification

The quantification of dNTPs from primary immune cells was performed by quantitative LC-MS. To remove the signal depression by the abundant rNTPs (*Kennedy et al., 2010*), a boronic acid chromatography step was performed prior to analysis. Briefly, NTPs were methanol extracted for 1 hr at 4°C from 5 million cells. Samples were supplemented with isotopically labeled (*Mansky et al., 2000*) C (*Priet et al., 2005*) N-dNTPs (Sigma-Aldrich) for accurate determination of each metabolite. Supernatants were dried under vacuum and then resuspended in 50 µL (200 mM Ammonium acetate, pH 8.8). This was directly applied to 100 µL of pelleted boronic acid resin and incubated at 4°C for 1 hr. Resin was pelleted by centrifugation, supernatant removed and dried under vacuum. Sample was resuspended in mobile phase A [2 mM $NH_4PO_4H_2$, 3 mM Hexylamine (HMA) in dd $H_2O$]. Chromatographic separation was performed with an analytical Hypersil Gold C18 column (100 × 1 mm, 3 µm particle size, Thermo Scientific, Waltham, MA), using a (Agilent, Santa CLara, CA) LC-MS system equipped with a binary pump. Mobile phase A was [2 mM $NH_4PO_4H_2$, 3 mM Hexylamine (HMA) in dd $H_2O$] and mobile phase B [Acetonitrile]. The flow rate was maintained at 50 µL/min and an injection volume of 20 µL. The autosampler was held constant at 4°C and the column at 30°C. The chromatographic gradient began at 10% mobile phase, followed by a linear gradient that reached 60% in 15 min. At the end of each run, the column was equilibrated for 10 min. Analyte detection was performed with the same LC-MS system with an electrospray ionization source, using multiple-reaction monitoring (MRM) analysis in positive ionization mode. The following MRM transitions (parent → product) were detected as previously described with the addition of dUTP: dATP (492 *m/z* → 136 *m/z*), dGTP (508 *m/z* → 152 *m/z*), dCTP (468 *m/z* → 112 *m/z*), TTP (483 *m/z* → 81 *m/z*) and dUTP (469 *m/z* → 81 *m/z*) (*Fromentin et al., 2010*). A standard curve was constructed using equimolar amounts of dNTPs.

## In vitro infection of primary immune cells

Infections with HIV viral strains (HIV[NL4.3(VSVG)], HIV[SF162(CCR5)], HIV[BAL(CCR5)]) were performed as previously described (*Jordan et al., 2003*). Briefly, primary cells were plated at a density of ~100,000 cells per well in a 96-well plate. Virus was added to each well, and the plate was spin infected for 2 hr at 1200 × g and 30°C and then incubated at 37°C until the given time points. Following incubation, cells were collected and stored at −80°C.

## Western blots for detection of APOBEC3A and APOBEC3G

We determined that A3G and A3A were poorly expressed or undetectable in MDMs prior to and after infection with HIV[SF162(CCR5)] (*Figure 2—figure supplement 1*). To validate the antibody activities and specificities, A3G and to a lesser extent A3A, were both detected in PBMCs, but not HEK 293T cells which are known not to express either enzyme (*Figure 2—figure supplement 1*). Protein extraction was performed using CelLytic M reagent (Sigma) according to the manufacturer's instructions. Protein concentration was determined by the Bradford assay (BioRad). Twenty-five micrograms of cell lysate was run on an SDS/PAGE gel and transferred to a PVDF membrane. Primary antibody (1:1000) was used for detection of APOBEC3G and is cross-reactive with APOBEC3A. Membrane was incubated overnight with anti-APOBEC3G obtained from the NIH AIDS Reagent Program, Division of AIDS, NIAID, NIH (cat. no. 10201; from Immunodiagnostics: Murine Human APOBEC3G (CEM15) Monoclonal Antibody (RRID:AB_2617158). The secondary HRP-conjugated anti-rabbit IgG (1:2000) was used for detection (ab97080; Abcam, Cambridge, MA,RRID:AB_10679808). Global protein expression was assessed by blotting MDM extracts for GAPDH (see below for further information).

## Western blots for detection of *vpr* in mutant virus

The primary antibody for GAPDH (1:2000) was a rabbit polyclonal antibody (ab15246; Abcam, RRID: AB_613387), and the secondary antibody (1:5000) was HRP-conjugated anti-rabbit IgG (ab97080; Abcam). Ten micrograms of each sample was run on an SDS/PAGE gel and transferred to a PVDF membrane. Detection of virion-associated viral protein R (vpr) was achieved using a rabbit polyclonal antibody (1:2000) obtained through the NIH AIDS Reagent Program, Division of AIDS, NIAID, NIH (cat. no.11836; from Dr. Jeffrey Kopp, RRID:AB_2617159) and secondary antibody (1:5000) was HRP-conjugated goat anti-mouse IgG (ab97040; Abcam, RRID:AB_10698223). Virion associated proteins were probed by re-suspending purified virus directly in buffer containing SDS. For an internal control, the same membrane was also blotted for p24 using the mouse monoclonal anti-p24 (1:2000) primary antibody obtained from the NIH AIDS Reagent Program, Division of AIDS, NIAID, NIH (cat. no. 24–2; from Dr. Michael H. Malim, RRID:AB_1124906). The secondary antibody (1:5000) used was an HRP-conjugated goat anti-mouse IgG (ab97040; Abcam).

## Excision droplet digital PCR (Ex-ddPCR) method and determination of the fraction of uracil in HIV DNA

To determine the uracil status of viral DNA, a modified droplet digital PCR (Ex-ddPCR) method was developed by inserting a pre-digestion step using UNG (*Figure 1*). Total genomic DNA was isolated using a QIamp DNA extraction kit (Qiagen) according to manufacturer's protocol. Genomic DNA was first fragmented using the endonuclease BSAJ-1 (1 U) in Cutsmart buffer (NEB) for 1 hr at 60°C. Uracil containing DNA sequences were digested using 50 nM UNG in 1x TMNB+ buffer (10 mM Tris-HCl (pH 8.0) 20 mM NaCl, 11 mM MgCl$_2$ and 0.002% Brij-35). Reactions were cleaned up using a MinElute PCR Purification kit (Qiagen). UNG or mock digested DNA were diluted in to a PCR master mix containing (1x Platinum taq buffer, 1x droplet stabilizer (RainDance Technologies, Billerica, MA), 1 mM dNTPs, 0.9 μM forward and reverse primers, 0.15 μM probe, 2 mM MgCl$_2$, 0.5 μM carboxy-X-rhodamine (ROX, Sigma) as a passive reference dye, 1 U platinum taq polymerase and varying amounts of template DNA and loaded in to a RainDance source chip (RainDance Technologies). Different primer/probe sets were designed to tile the HIV genome to report on heterogeneity of uracil incorporation. Amplification was performed with the following thermal program: 95°C for 10 min, followed by 44 cycles of: 95°C for 15 s and then 60°C for 1 min, followed by a final step of heating to 95°C for 10 min. The final high-temperature cycle cures the droplets. For each step, the ramp cycle time was decreased to 0.5°C/s. Amplification samples were transferred to a RainDance Sense instrument. Copy numbers of the human RNase P (*RPP30*) gene were measured in the same reaction mixture to determine cell number. DNA isolated from the Jurkat-based cell line (J-Lat) (*Jordan et al., 2003*) containing a full-length integrated HIV genome was used to establish gating parameters for HIV and RPP30-positive droplets. J-Lat cells were obtained through the NIH AIDS Reagent Program, Division of AIDS, NIAID, NIH: J-Lat Full Length Clone (clone 9846) from Dr. Eric Verdin. Ex-ddPCR data were analyzed using the RainDrop Analyst software package (RainDance Technologies). Drop-size gating was performed to remove drops of atypical size using a lower and upper size gate boundary for each PMT of 800 and 1150, respectively. Intact drops for each of the three clusters (one negative; quenched drops, and two positive; PMT1+ and PMT2+) were gated independently using the ellipse-gating tool. Gated clusters were spectrally compensated using the following definitions: (intact drops/quenched drops) as the negative control for both PMT1+ and PMT2+, while (intact drops/PMT1+) and (intact drops/PMT2+) were selected for PMT1+ and PMT2+, respectively. Positive cluster gating on spectrally compensated data was again achieved by using the ellipse-gating tool. A Poisson correction statistic was applied to both positive targets (HIV and *RPP30*) to remove sampling error (*Sanders et al., 2011*). Determination of uracil-free DNA transcripts was determined using *Equation 1*.

$$\text{FracU}^{\text{DNA}} = \frac{\{\text{positive droplets(no UNG)} - \text{positive droplets(+UNG)}\}}{\{\text{positive droplets(no UNG)}\}} \tag{1}$$

## Excision sequencing (Ex-seq) library preparation

Ex-seq libraries were constructed from genomic DNA by shearing for 12 min on BioRuptor (Diagenode, Denville, NJ), end repair, dA-tailing, and adapter ligation (NEB, Ipswich, MA) with dual indexed adapters (GSE76091). Library concentrations were quantitated with a Qubit fluorometer

and mixed at equal proportions to yield 500 ng of library DNA. HIV enrichment was performed using lockdown probes designed against the forward and reverse strand of the HIV construct used for infection, HIV$^{NL4.3}$ (GSE76091). Briefly, libraries were mixed with 5 µg Cot-1 and XGen Blocking oligos and hybridized with the lockdown probes at 3 pmol total for 4 hr at 65°C. Probes were isolated on Streptavidin Dynabeads (M-270; Invitrogen) to capture HIV DNA using Nimblegen SeqCap EZ Hybridization and Wash kit as indicated in the XGen lockdown probe protocol. DNA was eluted in water and samples were split in half, followed by uracil removal in one sample with UNG (5 U) and T4 Endonuclease IV (10 U) digestion for 2 hr at 37°C on beads. On-bead PCR was performed for 22 cycles with short Illumina primers (GSE76091) and Maxima Taq DNA polymerase (Invitrogen). PCR products were purified with 1.9 × volumes of Ampure XP beads (Beckman Coulter, Brea, CA) and quantified with a Qubit fluorometer. Forward and reverse lockdown reactions were pooled for UNG- and UNG+ independently and two lanes of Illumina HiSeq were performed. Data files from this study have been deposited to the NCBI Gene Expression Omnibus (GEO; [*Edgar et al., 2002*]) under accession GSE76091.

## Excision-seq data analysis

Demultiplexed FASTQ records were trimmed to remove low-quality cycles (cycles 1 and 2), and processed records were aligned to the human (hg19) and HIV reference (HIV$^{NL4.3(VSVG)}$) with bwa mem. Alignments in BAM format were converted to bedGraph coverage with BEDTools. Samples were deduplicated with samtools rmdup and Piccard markduplicates. VCF files were generated for mutation analysis from the rmdup files with freebayes and filtered with vcffilter for quality scores greater than 40 (*Garrison and Marth, 2012*). Mutations were further filtered for a depth of greater than 10 reads and 10 or more unique sequencing start sites. Chimeric and discordant reads were identified with samtools and were normalized to counts per million reads that mapped to HIV. A reproducible software pipeline for analysis of Excision-seq data is available at https://github.com/hesselberthlab/stivers-hiv.

## Ex-*ALU-gag* nested PCR for uracil detection

To specifically measure fractional uracilation of proviral DNA, we used a modification of the *ALU-gag* nested qPCR approach, in which an UNG digestion step is inserted before the initial PCR amplification (Ex-*ALU-gag* nested qPCR) (*Weil et al., 2013*; *O'Doherty et al., 2002*). Ex-Alu-*gag* nested PCR begins with the first amplification (40 cycles) used a primer complementary to genomic *ALU* sequences and a *gag* primer and ~100 ng genomic DNA. In addition to samples, reactions containing: *gag* primer only, no-template, and genomic DNA isolated from uninfected matched cells were used as controls. A standard curve for Ex-*ALU-gag* nested qPCR was generated using DNA isolated from J-Lat cells. Although there is some debate concerning the measurement of absolute proviral copy numbers using *ALU-gag* nested qPCR, Ex-*ALU-gag* nested qPCR is unambiguous with respect to determining the change in copy number resulting from UNG digestion. This is due to the fact that identical DNA samples are used for the PCR amplification steps. Thus, the fraction of the proviral population that contains uracil within the *gag* amplicon may be reliably measured.

## Cytokine stimulation of cultured MDMs

FACS sorted (GFP−/+) MDM populations were maintained in culture media for indicated times prior to cytokine stimulation. MDMs were either stimulated by co-culture with the IFNγ (50 ng/mL, R&D Systems, Minneapolis, CA) and 200 ng HIV$_{tat}$ peptide (ProSpec Ltd., East Brunswick, NJ) or IL-4 (50 ng/mL, R&D Systems). At 3 days post-stimulation (dps), the cytokine containing media is removed and replaced with fresh culture media. Viral supernatants were collected at varying time points from each well and assayed for HIV-1 p24 production using a sensitive ELISA (Alliance HIV-1 p24 antigen ELISA kit, Perkin Elmer, Hopkinton, MA).

## In vitro infection and nested limiting dilution DNA sequencing of CD4+ T cells

T cells were infected with replication-competent HIV$^{BAL(CCR5)}$ as previously described above. The HIV$^{BAL(CCR5)}$ virus was obtained through NIH AIDS Reagent Program, Division of AIDS, NIAID, NIH (cat. no. 510; contributors: Dr. Suzanne Gartner, Dr. Mikulas Popovic and Dr. Robert Gallo). Briefly,

CD4+ T cells isolated from PBMCs of an HIV-1-negative donor were purified using a negative selection method (CD4[+] T cell Isolation Kit II, Miltenyi Biotec) and activated with PHA for 3 days in IL-2-containing medium. A spinoculation method (2 hr, 37°C, 1200 × g) was used to infect activated CD4 + T cells with replication-competent HIV[BaL(CCR5)] virus (300 ng p24/10[6] cells). Following incubation for 6 days in IL-2 containing media, supernatants were collected and stored at −80°C. Freshly isolated CD4+ T cells were activated by αCD3/αCD28 stimulation for 3 days in IL-2 containing medium and then plated (100 μL/well) into a 96-well v-bottom plate. HIV[BaL(CCR5)] (500 ng p24/10[6] cells) was added to each well and cells were infected by spinoculation. Following infection, cells were suspended in 40 mL of IL-2 containing medium supplemented with enfuvirtide (10 μM) to prevent additional rounds of viral replication. Following a 2-day incubation at 37°C, genomic DNA was extracted from 10[7] CD4[+] cells using the Qiagen Gentra Purgene Cell Kit A. DNA was subjected to a nested limiting dilution PCR protocol adapted from *Ho et al.* with the following modifications: 30 cycles were run for the outer PCR, 40 cycles were run for each inner PCR, and all clonal outer wells were subjected to the 4 inner PCRs (A–D; corresponding to gag, pol, rev and env regions) regardless if positive or negative for the gag inner PCR (*Ho et al., 2013*).

## Extracellular viral RNA (EVR)-limiting dilution sequencing

HIV infected (HIV[SF162(CCR5)]) MDMs were sorted into GFP[−] and GFP[+] populations and diluted serially in five-fold increments (1× 10[5]–100 cells) using 12-well plates (Costar). Cells were maintained in culture for 14 days prior to cytokine simulation and supernatant collection. Extracellular viral RNA was collected from the supernatants of individual wells that contained the fewest number of input cells and were p24-positive in an attempt to obtain virus released from a cell containing a single provirus. Extracellular viral RNA was purified using a ZR-96 Viral RNA Kit (Zymo Research). Isolated viral RNAs were DNase-treated (Life Technologies) and reverse transcribed using a qScript cDNA synthesis kit (Quanta Biosciences) according to the manufacturer's protocol. Using the products of RT-PCR, nested PCR reactions were used to amplify the LTR and *env* regions of interest (primer sequences are listed in *Supplementary file 3*). Limiting dilution PCR was performed using Platinum Taq High-Fidelity polymerase (Invitrogen). For LTR amplification, outer primers 5LTROut and mod_VQA_R and inner primers 5LTRIn and mod_VQA_R were used. Thermocycler settings were the same for both outer and inner LTR PCR reactions: 94°C for 2 min, followed by 29 cycles of: 94°C for 15 s,55°C for 30 s and then 68°C for 1 min, followed by a final extension at 68°C for 7 min. For *env* amplification, outer primers ES7 and ES8 and inner primers Nesty8 and DLoop were used. Thermocycler settings were the same for both outer and inner *env* PCR reactions: 94°C for 3 min, followed by 25 cycles of: 94°C for 30 s, 55°C for 30 s and then 68°C for 1 min, followed by a final extension at 68°C for 5 min.

cDNA was diluted (1/20–1/1000) and used as input for the outer PCR reaction. Aliquots (1 μL) from each outer PCR product were used as input for inner PCR reactions and subjected to 1% agarose gel electrophoresis. Inner clonal PCR reactions from selected dilutions were used to identify dilutions where <20% of the PCR reactions were positive, where the corresponding outer PCR dilution contained one template with >90% of possibility by Poisson statistics. PCR products were gel extracted using the QIAquick Gel Extraction Kit (Qiagen) and directly sequenced (Sanger) without cloning (Genewiz, South Plainfield, NJ). Forward and reverse sequences for each sample were aligned into one consensus contig per sample using default assembly parameters with CodonCode Sequence assembly and Alignment software and aligned to HIV[SF162(CCR5)] reference sequence. Chromatograms showing double peaks were taken as evidence for more than one template present in the initial PCR reaction and were discarded from further analysis.

## Bronchio-alveolar lavage (BAL) cell processing

Bronchoscopy and broncho-alveolar lavage was performed on a single HIV-1 infected patient pre and post-ART treatment. At the time of the first bronchoscopy, the patient was ART naive with recent plasma viral load 24,162 copies/mL and CD4 459 cells/mL. Second bronchoscopy occurred 9.2 months later, after 6 months of ART treatment (efaviernz/emtricitabine/tenofovir) with plasma viral load <20 copies/mL and CD4 413 cells/mL. To obtain alveolar macrophages (AMs), bronchoscopy and lavage were done, as described elsewhere (*Popesu, 2014*). Briefly, lavage fluid was collected, filtered and BAL cells were pelleted by centrifugation. BAL cells were washed with HBSS and counted. After centrifugation, cells were re-suspended in Recovery Cell Culture freezing media

according to the BAL count ($10^6$ cells/mL) and stored in a −140°C freezer. PBMCs were isolated on a ficoll-hypaque gradient and were processed in the same manner as BAL cells.

## Determination of HIV *pol* copies and uracilation in in vivo infected monocytes and resting T cells using ddPCR and Ex-ddPCR

Monocytes and resting CD4+ T cells purified via negative selection (Miltenyi Biotec) from blood collected from six HIV infected patients on antiretroviral therapy (ART). Cell purity was determined as described above in *Purity of isolated cell populations*. Cellular DNA was extracted using a QIamp DNA Midi Kit (Qiagen) following the manufacturer's protocol. The frequency of total HIV DNA (*pol* copies/$10^6$ cells) was determined by ddPCR using published primers to conserved regions of HIV *pol* (*Strain, 2013*) and the reference cellular gene RNase P (*RPP30*) for genomic quantification. Total cellular DNA was estimated by halving the number of *RPP30* copies, and HIV copy numbers per diploid cell were calculated as the ratio of template *pol* copies per diploid cell (*pol* copies/$10^6$ cells). Due to the low frequency of HIV DNA in a large background of host cellular DNA, it was desirable to load as much total DNA (~$10^6$ cells) as possible to maximize assay sensitivity. See *Figure 7—figure supplement 1* for further details.

The limit of detection and quantification of the ddPCR assay was evaluated by establishing the loading limit and intrinsic limit of detection. A dilution series of uninfected PBMC DNA was used to determine the maximum amount of total DNA that could be accurately measured in a single ddPCR reaction. Cellular *RPP30* was used to monitor input DNA and showed a linear response across a wide range ($10^3$–$10^6$) of *RPP30* copies. The intrinsic limit of detection of (5 copies/$10^6$ cells) was determined by serially diluting DNA from infected CD4+ T cells into a background of healthy donor PBMC DNA (250,000 cells). For patient samples, replicate measurements were taken to improve the accuracy.

## Statistical analyses

Data were analyzed for statistical significance (HIV copy number and Frac $U^{DNA}$) by a two-tailed Student's t test for independent samples using GraphPad Prism (La Jolla, CA) using significance levels: \*\*\*p<0.001, \*\*p<0.01, \*p<0.05, not significant [ns]; n indicates the number of independent experimental replicates.

## Data access

Raw and processed sequencing data files (FASTAQ) from this study have been deposited to the NCBI Gene Expression Omnibus (GEO; http://www.ncbi.nlm.nih.gov/geo/) (*Barrett et al., 2011*) under accession number GSE76091. A reproducible software pipeline for analysis of Excision-seq data is available at https://github.com/hesselberthlab/stivers-hiv.

## Acknowledgements

We thank Dr. Amanda Brown for the replication competent GFP tagged viral plasmid (pSF162R3-GFP-Nef[+], HIV[SF162]), Dr. Matt Weitzman and Dr. Rahul Kohli for APOBEC3A/G antibodies, Dr. Janice Clements for the CEMx174 cells, and Kayla Herne for patient recruitment. We also thank the NIH AIDS Reagent Program, Division of AIDS, NIAID, NIH for critical reagents and all the study participants. This work was supported by the Martin Delaney CARE and DARE Collaboratories AI096113 and 1U19AI096109 (RFS), by the Johns Hopkins Center for AIDS Research P30AI094189 (RFS), by US National Institute of Health grants 43222 (RFS), NIAID R21AI112351 (JTS, JMS, JRH), NIAID RO1AI124777 (JTS), NIGMS RO1-GM056834 (JTS, JRH), NHLBI U01HL121814 (MBD), T32GM080189 (ECH), amFAR Research Grant 108834-55-RGRL (JTS, JMS, JRH), amFAR Innovation Grant 109361-59-RGRL (JTS), an ARCHIE Collaborative Research Grant from the Foundation for AIDS Research 108165-50-RGRL (R.FS.), Research Scholar Grant RSG-13-216-01-DMC from the American Cancer Society (JRH), and by the Howard Hughes Medical Institute and the Bill and Melinda Gates Foundation (RFS).

# Additional information

## Funding

| Funder | Grant reference number | Author |
| --- | --- | --- |
| National Institutes of Health | Chemical Biology Interface Training Grant T32GM080189 | Erik C Hansen |
| amfAR, The Foundation for AIDS Research | 108834-55-RGRL | Jay R Hesselberth Janet M Siliciano James T Stivers |
| National Institute of General Medical Sciences | RO1-GM056834 | Jay R Hesselberth James T Stivers |
| American Cancer Society | Research Scholar Grant RSG-13-216-01-DMC | Jay R Hesselberth |
| National Institute of Allergy and Infectious Diseases | R21AI112351 | Jay R Hesselberth Janet M Siliciano James T Stivers |
| National Heart, Lung, and Blood Institute | U01HL121814 | Michael Bradley Drummond |
| National Institute of Allergy and Infectious Diseases | RO1AI124777 | Janet M Siliciano James T Stivers |
| National Institutes of Health | 43222 | Robert Siliciano |
| Delaney AIDS Research Enterprise | AI096113 | Robert Siliciano |
| Howard Hughes Medical Institute | | Robert Siliciano |
| Bill and Melinda Gates Foundation | | Robert Siliciano |
| amfAR, The Foundation for AIDS Research | ARCHIE Collaborative Research Grant, 08165-50-RGRL | Robert Siliciano |
| Delaney AIDS Research Enterprise | 1U19AI096109 | Robert Siliciano |
| amfAR, The Foundation for AIDS Research | 109361-59-RGRL | James T Stivers |

The funders had no role in study design, data collection and interpretation, or the decision to submit the work for publication.

## Author contributions

ECH, MR, JRH, NNH, Conception and design, Acquisition of data, Analysis and interpretation of data, Drafting or revising the article; AAC, HZ, Acquisition of data, Analysis and interpretation of data, Drafting or revising the article; KMB, RAP, Conception and design, Acquisition of data, Analysis and interpretation of data; MBD, Conception and design, Drafting or revising the article, Contributed unpublished essential data or reagents; JMS, RS, JTS, Conception and design, Analysis and interpretation of data, Drafting or revising the article

## Author ORCIDs

James T Stivers, http://orcid.org/0000-0003-2572-7807

## Ethics

Human subjects: This study was approved by the Johns Hopkins Institutional Review Board and written informed consent was provided by both HIV-infected and healthy individuals (IRB00038590).

## Additional files

### Supplementary files

• Supplementary file 1. Mutational analysis of extracellular viral RNA produced from GFP$^-$ and GFP$^+$ MDMs. Infected MDMs containing proviruses were sorted into two populations according to GFP fluorescence at 14 days post infection and then activated with the indicated cytokines. Mutation frequencies are calculated based on total nucleotides sequenced (i.e. frequency = mutated nucleotide count/total nucleotides). RT-PCR was used to amplify extracellular viral RNAs into cDNA prior to clonal sequencing.

• Supplementary file 2. Mutational analysis of proviral DNA isolated from MDMsand CD4+ T cells. Infected MDMs containing proviruses were sorted into two populations according to GFP fluorescence at 14 days post infection and then activated with the indicated cytokines. Mutation frequencies are calculated based on total nucleotides sequenced (i.e. frequency = mutated nucleotide count/total nucleotides). In vitro infected resting CD4+ T cells. $^c$Insertion and deletions counted on a per event basis.

• Supplementary file 3. Primer and molecular beacon probe sequences (5'→3').

• Supplementary file 4. Patient demographics. Abbreviations: M; Male, F; Female, ABC; abacavir, DTG; dolutegravir, 3TC; lamivudine, DRV; darunavir, FTC; emtricitabine, r; ritonavir, ATV; atazanavir, TDF; tenofovir disoproxil fumarate, EVG/c; elvitegravir/cobicistat, EFV; efavirenz, N/A; not applicable.

### Major datasets

The following dataset was generated:

| Author(s) | Year | Dataset title | Dataset URL | Database, license, and accessibility information |
|---|---|---|---|---|
| Ransom M, Hesselbert JR | 2016 | Diverse fates of uracilated HIV DNA during infection of myeloid lineage cells | http://www.ncbi.nlm.nih.gov/geo/query/acc.cgi?acc=GSE76091 | Publicly available at the NCBI Gene Expression Omnibus (accession no. GSE76091) |

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
