## [Decision Letter]

Thank you for submitting your article "Diverse fates of uracilated HIV-1 DNA during infection of myeloid lineage cells" for consideration by *eLife*. Your article has been reviewed by three peer reviewers, one of whom, Stephen Goff, is a member of our Board of Reviewing Editors, and the evaluation has been overseen by Arup Chakraborty as the Senior Editor.

The reviewers have discussed the reviews with one another and the Reviewing Editor has drafted this decision to help you prepare a revised submission.

Summary:

This paper reports a set of observations suggesting that in MDMs, where dNTPs are low and dUTP unusually high, dUs are incorporated in place of dTs, with a variety of effects. There is a major restriction, involving UBER, in levels of proviral DNAs that are formed (only 1% of normal levels get through the restriction); and second, there is a high level of mutation in the resulting RNAs that are formed from the U-containing but not mutated proviruses, for obscure reasons (presumably during forward transcription). This last point is intriguing if mysterious.

The authors indicate that they have discovered a "new" mechanism, but there is a 25 year old literature describing UBER involvement in this process, due to high concentrations of dUTP in the non-dividing cells and lack of dUTPase to control the dUTP pool levels. The authors cite their own works as defining these parameters and have totally ignored the original publications, primarily demonstrated in animal models including FIV, EIAV, visna and CaEV.

The authors also indicate that they have identified two separate populations of MDMs in their studies, one permissive and one restricted for HIV infection and that one population Is exclusive in resulting in high level uracilation of viral DNA. As it turns out, these two populations are defined by the artificial expression of GFP, a marker of transfection for the viral genome(s) introduced; i.e., the GFP+ population expressed 7x the level of virus and virus-encoded products relative to the GFP- (essentially low or non-transfected) cell population. The authors never really address the differences in these populations other than obvious susceptibility to transfection,virus expression and relative uracilation. It seems probable that the GFP+ cells were more susceptible to transfection due to higher metabolic rate, most likely due to some stage of activation not extant in the GFP- population. The higher dUTP:dTTP ratio and lower virus integration/production would be consistent with this interpretation. It is unclear the extent to which uracil misincorporation might contribute to this, but seems unlikely to be the primary factor in low virus integration/expression.

They should also at least speculate on what that artificial i.e., defined by their cloning marker, population of monocytes represents. It must relate to the metabolic state of those cells and/or state of activation.

Essential revisions:

We need to have a better review of the history and the literature.

We need more discussion of the possible explanations for the two populations of expressing and nonexpressing cells – including some potentially trivial ones, as noted by one of the reviewers.

---

## [Author Response]

*We need to have a better review of the history and the literature.*

In response to a related editorial comment we now clarify our use of the word “new” with respect to this restriction pathway by modifying the final sentence of the abstract to read:

“These findings reveal new elements of a viral defense mechanism involving host UBER that may be relevant to the establishment and persistence of HIV-1 infection.”

We included a comprehensive review of the literature in our previous PNAS publication that included references to 20 previous publications in this area (Weil, 2013). Due to space restrictions we elected to refer to our previous review in the Discussion of this manuscript (Second paragraph of Discussion section). However, we are pleased to re-introduce this interesting history in a more condensed form to provide the best context for the current work. We have now added a short section in the introduction, including ten additional references that better review the literature:

“A role for dUTP and the UBER enzyme uracil DNA glycosylase (hUNG2) in HIV-1 infection have been long debated (see reference for a succinct review). […]These intriguing prior observations have motivated our further studies into the role of UBER in HIV infection, which now establish a profoundly restrictive role and unexpected effects on viral mutagenesis.”

*We need more discussion of the possible explanations for the two populations of expressing and nonexpressing cells – including some potentially trivial ones, as noted by one of the reviewers.*

We are very interested in exploring the nature of these two populations. Thus, prior to the original submission we had performed multicolor flow cytometry experiments to explore whether the uracilation/GFP- phenotype was correlated with the CD14/CD16 activation status of the MDMs. We now include a description of these findings, which establish that the uracilation-status is not correlated with the polarization state of the MDMs (at least with respect to these markers). We are actively pursuing metabolic profiling of these cell types to further elucidate their differences. However, this extensive work is the basis for another independent manuscript.

We have added the following section”CD14/CD16 status of MDMs does not correlate with GFP expression” describing the data that allows us to eliminate these polarization states as markers for uracilation status. We also comment on the possibility of complex metabolic differences between the expressing and non-expressing cells:

“CD14/CD16 status of MDMs does not correlate with GFP expression. […] It is possible that the dominant GFP^-^ phenotype arises from a complex combination of effects related to the metabolic state or state of activation of these cells.”